

# A Radiative Transfer Module for Calculating Photolysis Rates and Solar Heating in Climate Models: Solar-J 7.5

Juno Hsu[1], Michael Prather[1], Philip Cameron Smith[2], Alex Veidenbaum[3] and Alex Nicolau[3]

[1] Department of Earth System Science, University of California Irvine
   [2] Lawrence Livermore National Laboratory
   [3] Department of Computer Science, University of California Irvine

*Correspondence to:* Juno Hsu (junoh@uci.edu)

**Abstract.** Solar-J is a comprehensive model for radiative transfer over the solar spectrum that addresses the needs of both photochemistry and solar heating in Earth system models. Solar-J includes an 8-stream scattering, plane-parallel radiative transfer solver with corrections for sphericity. It uses the scattering phase function of aerosols and clouds expanded to 8th order and thus makes no isotropic-equivalent approximations that are prevalent in most solar heating codes. It calculates both chemical photolysis rates and the absorption of sunlight
and thus the heating rates throughout the Earth's atmosphere. Solar-J is a spectral extension of Fast-J, a standard in many chemical models that calculates photolysis rates in the 0.18-0.85 µm region. For solar heating, Solar-J extends its calculation out to 12 µm using correlated-k gas absorption bins in the infrared from the shortwave Rapid Radiative Transfer Model for GCM applications (RRTMG-SW). Solar-J successfully matches RRTMG's atmospheric heating profile in a clear-sky, aerosol-free, tropical atmosphere. We compare both codes
in cloudy atmospheres with a liquid-water stratus cloud and an ice-crystal cirrus cloud. For the stratus cloud both models use the same physical properties, and we find a systematic low bias in the RRTMG-SW of about 3 % in planetary albedo across all solar zenith angles, caused by RRTMG-SW's 2-stream scattering. Discrepancies with the cirrus cloud using any of RRTMG's three different parameterizations are larger, less systematic, and occur throughout the atmosphere. Effectively, Solar-J has combined the best components of
RRTMG and Fast-J to build a high-fidelity module for the scattering and absorption of sunlight in the Earth's atmosphere, for which the three major components – wavelength integration, scattering, and averaging over cloud fields – all have comparably small errors. More accurate solutions come with increased computational costs, about 5x that of RRTMG, but there are options for reduced costs or computational acceleration that would bring costs down while maintaining balanced errors across components and improved fidelity.



# 1 Introduction

A major challenge in simulating the Earth's climate is the tracking of solar energy, its absorption and scattering within and reflection from the Earth system, in the presence of heterogeneously distributed clouds and aerosols. The fifth assessment of Intergovernmental Panel on Climate Change (IPCC, Chapter 7, Boucher et al., 2013) summarizes that the net radiative feedback due to all cloud types is likely to be positive but with large uncertainty, mostly attributed to the uncertain impact of warming on low clouds. The confidence in the aerosol-climate feedback, through both aerosol and cloud albedo, is even lower and the uncertainty is $\pm 0.2$ W m$^{-2}$ ºC$^{-1}$. The major modeling challenges naturally point to the sub-grid parameterizations of clouds and cloud-aerosol interactions in coarsely-gridded global models, and the IPCC reports have documented substantial developments in the modeling of the chemical-physical properties of aerosols and clouds (Boucher et al., 2013). In comparison, relatively little attention has been paid to improving the treatment of aerosol and cloud scattering in climate models. This is both surprising and not. Solutions of the radiative transfer (RT) equations in scattering media are well documented with numerous methods and readily available packages such as TUV (Tie et al., 2003; Palancar et al., 2011) and SCIATRAN (Rozanov et al., 2014); however, these more accurate reference codes have always been viewed as too computationally expensive. Thus, in terms of climate model development, this is a solved problem with little intellectual interest, but too onerous to improve, and thus low-order approximations remain in place.

We present here Solar-J version 7.5, a radiative transfer model based on the computationally optimized photolysis code Fast-J (Wild et al., 2000; Bian and Prather, 2002; Sovde et al. 2012; Sukhodolov et al., 2016). Although this is the first version of Solar-J, we retain the numbering of the released versions of the core photolysis code, Cloud-J (Prather, 2015). The accurate treatment of cloud and aerosol scattering has been an essential requirement for atmospheric chemistry modeling, and Fast-J or alternative models (fast-TUV, Tie et al., 2003) are used standardly in global chemistry models. Solar-J is an extension of Fast-J wavelength range (0.18-0.8 microns) out to 12 µm and includes an 8-stream scattering solution for the absorption and reflection of sunlight over the full spectrum. Scattering and absorption by large aerosols (dust) and clouds are important for heating rates at these longer wavelengths. The long-term goal is to develop Solar-J as a single module for climate models, being marginally more expensive in computation, but delivering photolysis rates and more accurate shortwave heating rates, particularly for aerosol and cloud radiative forcing.

As finer grid resolutions and massively parallel computing are being pursued to enable more realistic atmospheric interactions with the land, ocean and biosphere in climate modeling, the radiative transfer codes implemented in most of the global models remain in their simplest possible analytical form of 2-stream scattering. With this approximation, all upward and downward scattering occurs at a single angle, and the scattering must be treated as isotropic, i.e., independent of sun angle. The ubiquitous adoption of 2-stream RT codes by the global climate and weather-forecasting models (e.g., DOE's Accelerated Climate Modeling for Energy (ACME), NCAR's Community Earth System Model (CESM), the European Centre for Medium-Range Weather Forecast (ECMWF) model) has been enabled by standardized packages like the Rapid Radiative Transfer Model for GCM Applications (RRTMG), developed based on the correlated-k approach (Mlawer et al., 1997; Clough et al., 2005). A 2-stream model was certainly necessary at a time when the need for

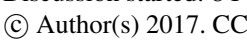



computational efficiency exceeded that for accuracy. With the rapid advancement of massive parallel
computing, it is time to ask if an upgrade to a higher-order scheme is needed for improved accuracy in climate
modeling, particularly with regard to cloud and aerosol forcing. The 2-stream scattering approximation has been
in use for decades in climate models and evaluating its systematic errors remains an active research topic (Li et
al., 2015; Barker et al., 2015). The errors are mostly from the inadequacy of using a single angle to represent

the scattering of cloud particles and aerosols. For example, the anisotropic, forward-peaked scattering of all
relevant atmospheric aerosols and cloud particles cannot be represented with the 2-stream approach, and all
scattering must be reduced to isotropic. To address this problem, a commonly used delta-scaling technique is
applied by removing the large forward-scattering peak, thus reducing the optical depth (Joseph et al., 1976;
Wiscombe, 1977). In addition, the Henyey-Greenstein (HG) phase function (Henyey and Greenstein, 1941) is

often used to tune the 2-stream scattering to better represent the scattering of large particles for specific sun
angles. Unfortunately, the HG phase function lacks the realistic back-scattering peak found for cloud particles,
particularly ice-crystals (Zhou and Yang, 2015). Li et al. (2015) find biases caused by the HG phase function
and conclude that higher-order moments of the phase function coupled with a multi-stream radiative transfer
algorithms are needed to improve accuracy. They demonstrate this point with a 4-stream δ-Eddington code

developed by Li and Ramaswamy (1996). Wild et al. (2000) tested the accuracy of different-order codes for
computing the mean radiation field in the presence of thick water clouds, and found that 8-streams were
necessary to have errors of only a few percent relative to a 160-stream code that resolved the scattering phase
function. For Solar-J, we adopt the Wild et al. (2000) optimization for water clouds and use Mie (liquid) or
Mishchenko (ice) (Mishchenko et al., 1996; 2004) full phase functions for scattering, truncate the expansion in

Legendre polynomials to order 8, and solve the scattering with 8 streams with no δ-scaling of the optical depth.

The Solar-J model and tests are described in Section 2. The resulting comparisons with RRTMG-SW are
presented in Section 3. Section 4 examines computational costs for Solar-J and options for optimization.
Conclusions and a path forward are discussed in Section 5.

## 95  2  Methods: model configuration and test cases

### 2.1  Solar-J spectral configuration

The 18 bins of Fast-J make up the first 18 bins of Solar-J and were optimized for calculating photolysis rates

below 64 km (Wild et al, 2000, Bian and Prather 2002). The first 11 bins (177-291 nm) are optimized around
the Schumann-Runge bands of $O_2$ and the Hartley bands of $O_3$, and the next 7 bins optimized for tropospheric
photolysis (291-850 nm). The bins were chosen to have relatively uniform opacities for the principal absorbing
species $O_2$ and $O_3$ across the wavelengths in each bin. In some cases, this includes combining different
wavelength regions on either side of the $O_3$ maximum cross section near 255 nm. Effectively, the 18 bins

extend the use of opacity distribution functions used to calculate $O_2$ photolysis rates in the Schumann-Runge
bands (Fang et al. 1974), an equivalent to the correlated-k method in the infrared (Lacis and Oinas 1991). An





inherent assumption is that any other scatterers and absorbers are uniform across each wavelength bin, justified by the narrowness of the bins and the lack of sharp spectral features in clouds and aerosols. Because Fast-J has been optimized against high-resolution spectral data for stratospheric ozone photolysis, and continually updated

with new cross sections (Sander et al. 2011), and tested against other codes (Palancar et al. 2011; PhotoComp (Eyring et al., 2010)), we have confidence in our stratospheric photolysis and heating rates.

The large bin 18 (412-778 nm) that includes the $O_3$ Chappuis band is unusual for Fast-J: it assumes a uniform absorption cross section for $O_3$, and it has a large factor-of-two change in wavelength. The $O_3$ cross sections vary smoothly over bin 18 and are $> 0.5 \times 10^{-21}$ cm$^2$ over the range 475-725nm with a broad maximum of $5\times10^{-21}$

cm$^2$ about 600 nm. Overhead opacity ranges from 0.4 to 4% over this band. With optically thin absorption, one can use the flux-weighted average cross section, $1.94\times10^{-21}$ cm$^2$, for the entire bin. Both the attenuation of sunlight and the absorption of photons to calculate the $O_3$ photolysis rate use this average. At very large air masses (solar zenith angles of 89-95 degrees) the atmospheric path approaches 1 optical depth and modest errors appear. If highly accurate calculation of the photolysis and heating rates due in the Chappuis band is required,

then further analysis of bin 18 is warranted, but otherwise this treatment is sufficiently accurate to follow these rates as the sun sets. Another possible source of error is that these cross sections are photon weighted, and for heating rates the cross sections should be energy weighted (Wm$^{-2}$). Fortunately, the energy-weighted $O_3$ cross section, $1.91\times10^{-21}$ cm$^2$, differs little from the photon-weighted one (with a result of $< 0.04$ K/day difference in clear-sky stratospheric heating).


RRTMG-SW has 9 large bins extending to wavelengths longer than the end point of Fast-J, and we adopt the flux-weighted average optical properties of clouds and aerosols for these bins as an extension to Fast-J/Cloud-J v7.3 to become Solar-J v7.5. Figure 1 shows the overlap of the spectral bins of Fast-J v7.3, Solar-J v7.5, and RRTMG-SW. Also shown is the revised Cloud-J v7.4 for which the long-wavelength edge of bin 18 has been

shortened from 850 nm to 778 nm to match the transition to RRTMG bins. The flux-weighted cross sections for several Fast-J species have been recalculated to account for this. Be aware that these rescaled cross sections apply to all Fast-J and Cloud-J versions 7.4 and later. Cloud-J remains a key component of Solar-J, as it produces representative samples of independent column atmospheres after considering the topology of cloud fractions (Prather, 2015). Solar-J has 27 major bins, referred to here as S-bins, e.g., S1-S27 in Table 4. Bins S1-

S17 are taken directly from Fast-J and have no sub-bins. The transition bin S18 combines Fast-J's uniform treatment of Chappuis-band $O_3$ absorption with 4 small non-overlapping sub-bins (17.5 out of a total of 608.7 Wm$^{-2}$) to include RRTMG's $H_2O$ and $O_2$ absorptions in their bins B24-B25. These four sub-bins have strong cross sections with their own distinct optical depth structures, and they do not overlap with the major $O_3$ absorption in bin S18. The rest of the non-ozone sub-bins (weak cross sections) are lumped into one sub-bin

and added to Solar-J's Chappuis band. In all, we take RRTMG's 14 sub-bins and optimized these to 5 total. The last 9 bins, S19-S27, are directly implemented from RRTMG and contain 78 sub-bins. The logic of having wavelength bins, and then sub-bins within them is to allow the gaseous absorbers with similar opacities to be gathered into one sub-bin, but to treat the scattering and absorption by aerosols and clouds as uniform across the major bin (see below). The fidelity of the spectral extension of Solar-J to match RRTMG is verified with the

clear-sky case presented in Section 3.1.





## 2.2 Clouds and aerosols

Like the photolysis rates calculated in Cloud-J, the heating rates in RRTMG-SW and Solar-J are highly sensitive

to the scattering and absorption from tropospheric and stratospheric aerosols, and from liquid-water and ice-water clouds. Cloud-J v7.4 has pre-computed tables of optical properties for typical aerosols and for both liquid- and ice-water clouds. For bins S1-S18, many of these are effectively non-absorbing. With the extension to longer wavelengths, it becomes important to treat the absorption by clouds and the stratospheric sulfate layer. We take the refractive indices for liquid water, ice water and sulfuric acid and calculate solar-flux weighted

mean values for each bin S12 – S27. Bins S1-S11 do not reach the troposphere in significant amounts and hence they just repeat the properties of bin S12. For the first 18 bins optical properties are weighted by the solar photon flux (photons cm$^{-2}$ s$^{-1}$), and the last 9 bins are weighted by the solar energy flux (Wm$^{-2}$). These refractive indices are combined with a Mie scattering code and a model for the size distribution of particles to calculate the effective radius ($r_e$), single scattering albedo (SSA), ratio of optical to geometric cross section (Q),

and the first 8 terms in the expansion of the scattering phase function ($A_{0:7}$) that includes the asymmetry parameter ($g = A_1/3$).

For liquid water we take the refractive index from FORTRAN codes developed at the U. Wisconsin Madison by M.A. Walters for liquid water (NDXWATER: Hale and Querry (1973); Palmer and Williams (1974); Downing and Williams 1975) and ice water (NDXICE, based on Warren (1984)). Liquid water clouds use Deirmendjian's

C.1 gamma distribution of drop sizes ($\alpha = 6$, see Deirmendjian 1969) and the Mie code from Hansen and Travis (Hansen; Travis 1974) for a range of effective radii: $r_e = 1.5, 3, 6, 12, 24, 48$ μm, see also Hess et al. (1998). Optical properties (SSA, Q, $A_{0:7}$, g) are calculated for bins S12-S27 for these effective radii and then individual cloud properties at each bin are interpolated piecewise linearly in $r_e$. Cloud properties for S1-S11, wavelengths where sunlight does not reach the troposphere, just take the values from S12.

For ice-water clouds we have two T-matrix computations supplied by M. Mishchencko for Fast-J (Mishchenko et al. 2004) for warm (irregular) and cold (hexagonal) ice clouds. These included Q and the scattering phase function (including $A_{0:7}$) for the visible region (~600 nm) and were used at all Fast-J wavelengths. When there is significant absorption the values of SSA, and to some extent Q, are complex functions of $r_e$ and do not simply scale as total mass. For this first version of Solar-J, we made a simplifying assumption and used the Mie code

with the ice-water refractive index to calculate SSA and Q as a function of $r_e = 3, 6, 12, 24, 48, 96$ μm using the liquid-water cloud's C.1 distribution. Effectively we assumed that the ice particles were spheres. As with liquid water, optical properties were calculated for S12-S27, and S1-S11 use S12. For the phase function $A_{0:7}$, we kept the two T-matrix results (irregular and hexagonal ice particles) and used them for all $r_e$ of that type of ice cloud. The obvious next upgrade to Solar-J is a redo of the ice-water clouds with a broader, better mix of cloud types

(Mishchenko et al. 2016; Yang et al. 2015).

The refractive index for mixtures of sulfuric acid and water are also well characterized (Beyer et al. 1996; Biermann et al. 2000; Krieger et al. 2000; Myhre et al. 2003), and we use the tables from Lund-Myhre et al (2003). For the stratospheric sulfate layer, we chose background and volcanic bimodal log-normal size distributions based on Deshler et al. (2003): background has a dominant mode (98%) with $r_e = 0.125$ μm and a

secondary mode with $r_e = 0.432$ μm for an average of $r_e = 0.131$ μm; volcanic has a dominant mode (81%) with





$r_e$ = 0.487 µm and a secondary mode with $r_e$ = 0.149 µm for an average of $r_e$ = 0.422 µm. The stratospheric aerosol properties are tabulated for bins S5-S27 for a combination of temperatures (220-250-280K) and weight-percent sulfuric acid (50-70-90%) with 220K and 70% being typical for the stratosphere (McGouldrick et al., 2011). The refractive indices and size distributions of tropospheric aerosols are not as well characterized. Fast-J has a collection of aerosol optical properties for wavelengths 300-800 nm based on community contributions (e.g., Liousse et al. 1996; Martin et al. 2003), and this has been propagated for testing in Solar-J. However, if heating by tropospheric aerosols such as brown and black carbon and dust is to be accurately modeled with Solar-J, then one must go to the specific models to acquire the physical and optical properties, e.g., NCAR's CESM 1.2 (Tilmes et al. 2015).

The Solar-J bins, solar fluxes ($S_{phot}$ in photons cm$^{-2}$s$^{-1}$ and $S_{Watt}$ in Wm$^{-2}$), and Rayleigh cross-sections ($X_{Rayl}$ cm$^2$) are summarized in Table 1. The spectral properties for examples of liquid-water clouds ($r_e$ = 12 µm), ice-water clouds ($r_e$ = 48 µm, cold, hexagonal), background stratospheric 70 wt% sulfuric acid aerosols, and volcanically enhanced stratospheric aerosols for each Solar-J bin are given in Table 2. This table gives wavelength data for the real and imaginary refractive indices based on the flux-weighted means, as well as the Mie-derived values for Q, SSA, and g. The relative importance of cloud heating in each bin can be estimated by multiplying the solar energy by the absorbing fraction, $S_{Watt}$ x (1 − SSA). One finds that absorption for bins S1-S20 is negligible, that both types of clouds and stratospheric sulfate aerosols have large absorption in bins S25-S27, and that ice-water clouds have large absorption per optical depth in bins S21-S24 while liquid-water clouds do not. Ice-water and liquid-water have real refractive indices that differ by at most 5%, and imaginary refractive indices that differ typically by a factor of 2 (except for S27). The cause of this difference in specific absorption is the ratio of mass (which controls absorption) to surface area (which controls optical depth), i.e., it is proportional to $r_e$.; and ice-water clouds typically have 4x greater $r_e$.

### 2.3 Test cases: clear-sky, clouds and the optical properties


To compare Solar-J and RRTMG, we adopt a standard atmospheric column model, typical of the tropical oceans (surface albedo = 0.06) and define three cases: clear sky, a stratus liquid-water cloud, and a cirrus ice-water cloud. Both cloudy cases assume 100% cloud cover; the cloud overlap algorithms of Cloud-J are not invoked. Neither are aerosols included. Atmosphere and cloud properties are given in Table 3. Each test case is evaluated at four different solar zenith angles (SZAs) at 0°, 21°, 62°, and 84°, whose respective cosine values are and 1.0, 0.93, 0.47 and 0.10.

The two cloud profiles are extracted from the 3-hourly, July 2005 ECWMF-Integrated Forecast System (IFS) data. This data set has a horizontal resolution of 1° x 1° in longitude and latitude and 37 vertical layers with about ∼½ km vertical resolution in the troposphere. Our example of marine stratus clouds has liquid water content (LWC, g m$^{-3}$) only below 2 km, while the cirrus example has non-zero ice water content (IWC, g m$^{-3}$) above 6 km and no liquid water anywhere. The total cloud water content (CWC, g m$^{-3}$) and effective radius ($r_e$) are also listed in Table 3. Solar-J has default values for $r_e$: for cirrus they are parameterized as $r_e$ = 164 x IWC$^{0.23}$ µm, based on a fit to the data in (Heymsfield et al. 2003); and for liquid-water clouds are based loosely



on observations of clean maritime stratus (Boers et al. 1996; Gerber 1996; Miles et al. 2000), with $r_e$ = 9.6

micron at pressures greater than 810 hPa and increasing linearly to 12.7 microns at 610 hPa and above.  When

implemented in an atmospheric model, $r_e$ will ideally be supplied by the atmospheric model driving Solar-J.

Heating rates and the changes in the radiative energy budget due to clouds are evaluated with the clear-sky

component subtracted.  In both Solar-J and RRTMG, when $r_e$ and CWC are given, the corresponding

wavelength-dependent properties are derived from tables or formulae.  In Solar-J the scattering phase function is

truncated at 8 terms, but in RRTMG's 2-stream model only the first term ($A_1/3$ = g) is retained.  For liquid

water, RRTMG adopts the parametrization scheme by Hu and Stamnes (1993). For ice clouds three different

parameterization are available, and all are tested here (Ebert and Curry, 1992, henceforth EC92; Key, 2002,

henceforth Key02; Fu, 1996, henceforth Fu96).

These parameterization schemes in RRTMG aim to fit the ice-cloud optical properties - extinction coefficient,

SSA and g - as a polynomial function of $r_e$ and CWC. Note that Fu's parameterization is based on the

generalized effective diameter ($D_{ge}$) but can be related to the input $r_e$ through Eq. 3.12 of Fu (1996). Elbert and

Curry's parameterization has been applied in the Community Atmosphere Model (CAM version 4.0 and prior

versions).  According the documentation in RRTMG, Key's parameterization was taken from the Mie-

calculated spherical shapes of ice particles from the Streamer radiative transfer codes (Key, 2002), and thus

should be similar to the Solar-J approximation.  The two-stream solution to the radiative transfer problem, as

implemented in RRTMG, requires that the scattering optical depth ($\tau_{scat}$) be reduced with what is described as

the δ-Eddington approximation (Huang 1968; Joseph et al., 1976).  The purpose is to remove the forward-

scattering peak typical of large particles and have only isotropic-equivalent scattering.  The absorption optical

depth is not changed to ensure correct absorption in the limit of optically thin clouds.  The basic problem with

these approximations is that the cloud optical depth is reduced by as much as a factor of five, and thus

substantially more sunlight is transmitted through the cloud as a direct solar beam rather than as scattered light.

In RRTMG (except for the Fu96 ice-cloud approximation) the Henyey-Greenstein (HG) phase function (Henyey

and Greenstein, 1941) is further used to approximate the scattering of aerosols and clouds because of its simple

power series formulation.  The HG phase function does not represent realistic scattering because it does not

have backward-scattering peak of real aerosols and clouds.  As might be expected, errors in two-stream

approximations are ubiquitous and vary widely with solar zenith angle (Boucher 1998).

### 3  Results:  Solar-J versus RRTMG

#### 3.1  Clear sky


The clear-sky comparison between Solar-J and RRTMG for overhead sun (SZA = 0°) is summarized in Table 4

and Figure 2. Table 4 lists the band-by-band radiation budget in Wm$^{-2}$, with Solar-J's spectral bins labelled as

S-bins and RRTMG's as B-bands (B16-B29 follow the same band numbers as in RRTMG's codes). For easy

comparison, several Solar-J's spectral bins of higher resolution from the UV range are lumped together to best

match the RRTMG's bin of similar range, and vice versa with RRTMG's B24 and B25 bins combined to





compare to Solar-J's S18 bin. The incoming spectral solar irradiance is slightly different for the two codes and so for easier comparison we scale each of them to a total of 1360.8 Wm$^{-2}$ (Kopp and Lean, 2011). RRTMG adopts the solar source function from Kurucz (1992), while Solar-J integrates high-resolution (0.05 nm) photon fluxes (Meier and Stamnes, 1992) by wavelength to obtain the solar irradiance. Clear-sky summary comparisons
for the other three SZAs (21°, 62°, 84°) are shown in Table 5 under Clear-Sky columns.

In Table 4, the incoming spectral solar irradiance at top of the atmosphere (TOA down) is balanced by components of (1) the reflected flux going back to space (TOA up positive), (2) the absorption in the atmosphere, separated into stratosphere and troposphere, and (3) surface heating. Several differences in the
configuration of spectral bands between Solar-J and RRTMG affect these results. For one, RRTMG does not include the small amount of solar irradiance at wavelengths ($\lambda$) < 200 nm (0.06 Wm$^{-2}$), and thus ignores photodissociation of $O_2$ molecules in the Schumann-Runge bands and part of the Herzberg continuum that heats the upper stratosphere and mesosphere. Second, for $\lambda$ =200-345 nm, Solar-J has 3 Wm$^{-2}$ (6%) less solar energy than RRTMG and the difference appears in RRTMG's larger heating of the stratosphere. Third, the bin division
between 345 and 778 nm is at 412 nm for Solar-J (i.e., between S17 and S18), but at 442 nm for RRTMG (between B26 and B24+B25). This interval, 412-442 nm has very low $O_3$ absorption, significant Rayleigh scattering, and a large amount of solar energy (~51 Wm$^{-2}$). Both the shorter-wavelength bins (S17 or B26) reflect about 20% of the incoming radiation, but in the adjacent bin with the Chappuis $O_3$ band it is only about 9%. Thus, placing the 412-442 nm interval with the Chappuis band results in greater atmospheric absorption
and less reflection. Solar-J (and Cloud-J) should investigate moving the band edge to 442 nm.

These differences, particularly the 412-442 nm interval, explain most of the total budget difference where, overall, Solar-J reflects 4 Wm$^{-2}$ (4%) less back to the space, absorbs 2 Wm$^{-2}$ (6%) less in the stratosphere, 3 W m$^{-2}$ (1%) more in the troposphere, and 4 Wm$^{-2}$ (1/2%) more at the surface. For SZA = 21° and 62° (Table 5), Solar-J continues to reflect 4 Wm$^{-2}$ less energy back to the space, but at large SZA= 84° the two models match
closely. While spherical effects may play some role in this shift, we suspect that Rayleigh scattering may contribute. The forward-backward enhancement in Rayleigh scattering is not represented in 2-stream isotropic scattering. Thus RRTM – Solar-J differences will shift as the primary beam shifts from vertical to horizontal as a much greater fraction of the visible light is scattered. At low sun the Rayleigh optical slant path along the solar beam is much greater than 1 for bin S17 and even ~1 for S18.

Figure 2 compares the vertical profiles of clear-sky heating rates for overhead sun (SZA = 0°) with the abscissa axis scaled separately for the stratosphere and the troposphere. Both models produce similar structures with the heating maximum in the stratosphere about at 45 km altitude and in the troposphere between 2 and 8 km. The unusual zig-zag structures of heating in the troposphere are unphysical and related to the approach of RRTMG and other correlated k-distribution approaches (Lacis and Oinas, 1991) in binning the line-by-line opacities for
the sub-bins. Instead of a continuum of water vapor opacities in a large bin, there are a discrete number of monotonically increasing cross sections for the sub-bins. The ability of Solar-J to match these structures demonstrates that Solar-J has correctly implemented the RRTMG spectral model. The consistent Solar-J minus RRTMG difference of 0.05 K/day near the surface in Figure 2d comes from Solar-J's simplification of combing RRTMG's 14 sub-bins with $O_2$ and $H_2O$ absorption in bins B24-B25 into the 5 sub-bins of S18. Solar-J minus





RRTMG differences are shown in the right two panels. In the troposphere these are small, but in the
       stratosphere there is a clear bias with Solar-J producing more heating above 40-50 km and less heating below.
       Differences at the top, above 50 km, are due in part to the lack of λ <200 nm radiation in RRTMG, and in part
       due to a better resolution of the $O_3$ and $O_2$ cross sections in Solar-J. Overall, RRTMG deposits more energy in
       the lower stratosphere, below 35 km, except at larger SZA where it deposits less. Thus, for the tropics and mid-
latitudes, RRTMG will overheat the lower stratosphere, possibly changing the stability and wave propagation to
       the high latitudes (Hsu et al. 2013). At high latitudes, RRTMG error is in the opposite direction, resulting in a
       colder polar stratosphere with possibly stronger winter vortices.

       Solar-J traces the solar beam through a spherical atmosphere back to the sun. RRTMG assumes a flat Earth.
       Both then calculate the subsequent scattering and absorption in a plane-parallel, flat atmosphere, but with
different solar source terms at each level. Solar-J is able to simulate both photolysis and heating rates
       throughout twilight, even when the sun is no longer directly visible at the layer. Figure 3a shows the smooth
       decline in $O_3$ photolysis rates as the SZA passes from 84° to 95°. Figure 3b shows the corresponding heating
       rates from both RRTMG and Solar-J. The lack of sphericity in RRTMG leads to large systematic negative
       biases in the heating rates at low sun. Sphericity errors extend up to SZA = 80° but are largest of course at
twilight. The high-latitude atmosphere will have SZA >80° for much of the day, and thus RRTMG may lead to
       a cold bias for the high latitudes.

### 3.2 Low-level marine stratus cloud

For the stratus cloud, the liquid water path (LWP, g m$^{-2}$) in each layer is derived from the LWC and height of
       each layer (Table 3) and is plotted vs. altitude in Figure 4a as described in Section 2.3. The resulting cloud
       optical depth in each layer, τ, (evaluated at 600 nm) is also written in pairs with Solar-J's as the first number and
       RRTMG's reduced delta-scaled optical depth (τ') as the second. Both RRTMG and Solar-J start with same
       value of τ because the Mie-based scattering phase functions for liquid water are unambiguous and both adopt
the same values for $r_e$, Q, and density of liquid water. The $r_e$ is set to 9.6 μm through most of this cloud profile.
       The LWC increases from the surface to a maximum of 0.12 g m$^{-3}$ at 1.25 km and falls off to zero by 2.3 km
       altitude. Because of the increasing thickness of the model layers with altitude, the LWP and layer τ are not as
       smoothly peaked. We deem this profile realistic from comparing to the observed range for coastal marine low
       clouds (see Figure 4 of Hu et al. (2007) for July liquid cloud radii distribution and Figure 1(a) of Painemal et al.
(2016) for LWP).

       Table 5 summarizes the clear-sky heating rates and the stratus cloud radiative effect (CRE, W m$^{-2}$, calculated as
       change relative to clear sky) for Solar-J and RRTMG for the four SZA used here. At overhead sun (SZA=0°)
       with the solar input at 1360.8 W m$^{-2}$, the effect of this low-level marine stratus cloud (per Solar-J) is to reflect
       an additional 469 W m$^{-2}$ back to the space, absorb an additional 91 W m$^{-2}$ in the atmosphere primarily within
the cloud, and thus to reduce the surface heating from 969 to 409 W m$^{-2}$. As in the clear-sky comparison, both
       models look broadly similar but with some large systematic biases. For SZA = 0-62°, Solar-J reflects ~10 W m$^{-2}$
       (2-3%) more sunlight back to space; both models calculate about the same increase in atmospheric absorption;



RRTMG consistently absorbs less energy within the cloud but more above it; and thus Solar-J calculates greater reduction in surface heating (also about 2-3%) than RRTMG. These differences in solar heating are large

compared with anthropogenic climate forcing from greenhouse gases (~4 W m$^{-2}$) (Myhre et al., 2013), but of course stratus clouds occupy only a fraction of the surface. Within the atmosphere, there is a large difference in the distribution of CRE, with Solar-J calculating 5% (SZA=0°) to 20% (SZA=62°) more in-cloud heating than RRTMG. The profile of heating rates (Figure 4b) shows a double peak at 1.9 km (visible τ ~ 1) and 1.2 km (τ ~ 6) even though the LWC has a smooth maximum at 1.1 km. The longer wavelength bins (S25-S27) are fully

absorbed in the uppermost part of the cloud (τ < 1), while the shorter wavelengths (S19-S24) penetrate the cloud to scattering optical depths of order τ ~ 8. RRTMG consistently calculates lower in-cloud rates, see below. It is possible that Solar-J's greater heating in stratus clouds may change the dynamics of stratus clouds relative to a model using RRTMG (Harrington et al., 2000). At low sun (SZA=84°) Solar-J calculates 4% greater reflectance change; both models calculate less atmospheric heating within the cloud but more heating above it;

and the surface heating in Solar-J is about 2 W m$^{-2}$ less than in RRTMG. Both models show enhanced heating only in the uppermost cloud layers above 1.7 km (Figure 4b).

We believe that the RRTMG biases identified here are errors caused by the 2-stream approximation. This is supported by the study of Li et al. (2015, see their Figure 2), who show small negative errors in absorption from the calculation of δ-Eddington (2-stream) approximation in the case of the single-layer liquid cloud ($r_e$ = 10 μm,

τ ~ 4) with cos(SZA) > 0.2 (i.e., our SZA = 0-62°). For our SZA=84° this absorption bias reverses as is also found by Li et al. (2015) for cos(SZA) <0.2. In their study the 2-stream calculations are compared to the 128-stream DISORT (Discrete-Ordinate) benchmark calculations using accurate phase functions and no δ-scaling (similar to the study of Wild et al., 2000). One source of error in RRTMG's model is the choice of δ-scaling factor, which they base on the HG phase function using only g. Alternatively, one can use the 2nd moment of

the true Mie phase function (Wiscombe, 1977). We revised the RRTMG code to do this using Solar-J's scattering phase functions and found a modest reduction in this error from -14 W m$^{-2}$ to -9 W m$^{-2}$ for reflected sunlight (SZA=0°).

### 3.3 Tropical cirrus cloud


For the cirrus cloud comparison, we use all three ice-water parameterization options in RRTMG and Solar-J's single parametrization. Figure 5ab shows the prescribed profiles of model input of IWC and $r_e$ (Table 3). The cumulative overhead τ at 600 nm is shown in Figure 5c. The δ-scaling varies considerably across the RRTMG parameterizations: Solar-J's unscaled τ ~ 0.43 compares with EC92's τ ~ 0.25, Fu96's τ ~ 0.15, Key02's τ ~

0.09 (see also Table 6). Thus, the fraction of sunlight scattered by the cloud varies widely across all four. The asymmetry parameter g from Mishchenko's phase functions for hexagonal and irregular ice used in Solar-J ranges from 0.75 to 0.81 (as compared to 0.88 for equivalent-size liquid-water clouds), but g values for all RRTMG ice clouds range from 0.4 to 0.6 for wavelengths where scattering is important (S12-S24). The absorbing optical depth, $τ_{abs}$, is a very important diagnostic because in an optically thin cloud the overall heating





should be proportional to it. Table 6 shows that all four ice cloud models have similar $\tau_{abs}$ up to S22, and if we average S23 and S24 (which appears to have been done in EC92), then all four models remain similar in terms of solar absorption. As noted for the stratus cloud, all models predict a large, factor of 5, jump in $\tau_{abs}$ for S25-S27 ($\lambda > 2.5$ µm), which are the most important bins for cirrus cloud heating. At these wavelengths, EC92 has the largest absorption $\tau_{abs}$, about 0.3, followed by Solar-J's 0.21.

Cloud heating rate profiles at SZA = 0° are shown in Figure 5d, and the large range clearly reflects the $\tau_{abs}$ for S25-S27. The cirrus CRE for four SZAs and for five components (reflected at top of atmosphere, absorbed in above-cloud atmosphere, in-cloud atmosphere, below-cloud atmosphere, and absorbed at surface) are shown as a set of 20 bar charts in Figure 6. The CRE percent changes relative to clear-sky are shown as four color bars representing Solar-J (red), EC92 (blue), Fu96 (green) and Key02 (yellow). The clear-sky energy flux (W m$^{-2}$)

averaged over the four models are shown in a larger font in each bar chart. For example, at SZA = 21° the energy absorbed by clear-sky atmosphere over the altitude range of the cirrus cloud is 112.8 W m$^{-2}$. The CRE in Wm$^{-2}$ within the cirrus cloud for Solar-J is then 112.8 x 8.8% (red bar) = + 9.9. The value of each bar (%) is also written out immediately above/below the bar in a small font. The y-axes in Figure 6 have different scales at different SZA.

A key cirrus CRE is the increase in albedo, the top-of-atmosphere reflected sunlight, as shown for all models and a range of SZAs in Figure 6 (top row). The percent increase across RRTMG models (13-122%) scales in proportion to $\tau$, with EC92 being the largest and Key02, the smallest. This relative order stays the same across all SZAs, but the range across RRTMG models decreases and the relative percent increases for larger SZA. The Solar-J model also increases in percent with SZA, but the pattern is different than for RRTMG models. At

overhead sun, Solar-J has about the same CRE percent as EC92 even though it has 1.7x greater $\tau$. This can be understood in that Solar-J cirrus is highly forward scattering and less of the scattered light is reflected backward and upward. As the SZA increases to 21-62°, however, the peak in backscatter at 180° becomes less important and Solar-J shifts lower relative to EC92 to look like Fu96. At very large SZA = 84°, with most of the sunlight being scattered at least once within the cloud, the Solar-J model again looks like the largest $\tau$, model EC92. To

first order the Solar-J model is calculating the correct SZA dependence of the CRE by using both a more realistic scattering phase function and 8-stream scattering. The use of Mishchenko's sample T-matrix phase function may not be a perfect choice for cirrus, but it is clearly more realistic than the isotropic scattering used in RRTMG. Solar-J captures the cirrus albedo curve similar to Figure 2 of Mishchenko et al. (1996) for $\tau = 0.1$ in which the slope increases rapidly as cosine (SZA) approaches to 0. While the RRTMG 2-stream models can

be tuned to be correct answer at some SZA, they will have errors of 15 W m$^{-2}$ at others. The change in surface heating (5$^{th}$ row) looks like the reverse of the top-of-atmosphere bars with similar relative weighting of the RRTMG models. Again, it shows that 2-stream scattering cannot mimic the correct SZA dependence of reduced surface heating under cirrus.

With greater reflection of sunlight, the atmospheric heating above the cloud increases in all cases. With

RRTMG the scattered light has only one angle, and thus the above-cloud heating (2$^{nd}$ row of Figure 6) is strictly proportional to the top of atmosphere increases. With Solar-J the reflected light is calculated at four zenith angles with the flux at larger zenith angles producing more heating (i.e., longer slant-path through the





atmosphere). This is most apparent in the SZA = 84º case where the low-angle scattering driven by the low solar elevation produces relatively much more atmospheric heating.

In-cloud heating (3rd row) is expected to be proportional to $\tau_{abs}$ at high sun (SZA = 0-62º), and for flux-weighted bins S25-S27 these $\tau_{abs}$ are 0.31 (EC92), 0.21 (Solar-J), 0.17 (Key02), and 0.16 (Fu96). While the actual heating of the cirrus ice particles may be in this proportionality, all we calculate is the total change of heating over the in-cloud layers. As seen in Figure 6 there is substantial clear-sky absorption by atmospheric water vapor in the cloudy layers (~100 W m$^{-2}$) at high sun. Thus, the small perturbation caused by the cloud (<10%, 3rd row) result

from in-cloud heating of ice particles (proportional to $\tau_{abs}$) countered by reduced heating of the water vapor in the region because of the increased upward scattered light (top row). The extreme case of SZA = 84º has all models calculating 20-40% reductions in heating because of the reduced sunlight. We can understand the erratic results of Fu96 (see Figure 5(d) green line and green bars for in-cloud absorption in Figure 6) in that this model's δ-scaling selects altitude-dependent g-values (as a function of ice crystal size) while both EC92 and

Key02 derive vertically uniform g-values from δ-scaling. Thus the Fu96 scattering within the cirrus cloud is very different from the other RRTMG models. Artificially fixing Fu96 g-values at a fixed mean value throughout the cirrus profile recovers a heating profile similar to Key02's in several bands where they have similar τ. The lesson here is that the scattering model is critical for calculating in-cloud heating, even for optically thin cirrus.

## 4 Computational costs

The major computational costs of Solar-J and similar codes within a chemistry-climate model centers on three key components: matrix operations required for multi-stream scattering; wavelength bins representing the spectrum of optical properties, and approximation of the multitude of independent column atmospheres (ICAs)

resulting from a complex overlapping cloud field within a grid cell. What is a reasonable requirement for multi-stream scattering in a climate model? From this work as well as a history of publications noted above, the analytic 2-stream approximation has errors that cannot simply be corrected or averaged over, that create large-scale biases in cloud radiative forcing with latitude, and that significantly misrepresent the direct:diffuse ratio of solar radiation at the surface. The original Fast-J work (Wild et al., 2000) examined a range of multi-stream

scattering models and found that for typical clouds, an 8-stream solution was able to match within a few percent that of a hundreds-stream code for the mean intensity above, within and below the cloud. A major advantage of 8-stream was that no δ-scaling is needed and a simply truncated scattering phase function can be used directly. The parent RRTM-SW code has the option of using a more accurate 16-stream scattering code, but would in general be computationally much more expensive than the Fast-J 8-stream. The basic costs of the matrix

inversions (Fast-J via Feautrier, 1964) or eigenvalue solutions (RRTM via DISORT, Stamnes et al., 1988) scale as $n^3$. For the same 8-stream solution, DISORT performs eigenvalue decomposition of 8x8 matrices at each level at a cost of order $8^3$, while the Feautrier solves the finite-difference equations with 4x4 matrices at split levels for a cost of order $2x4^3$. As a first guess the Feautrier code should run 4x faster than the equivalent DISORT code. We examine the costs and options of wavelength binning and cloud-field approximations

below.



### 4.1  Solar-J vs. RRMTG-SW

Cloud-J (and hence Fast-J) have been extensively tested in the UCI Irvine Chemistry-Transport Model (CTM).
Fast-J timings are estimated by comparing full cloud quadrature (2.75 calls per column atmosphere per time step, see below) versus an average-cloud approximation (1 call). We find that 12% of the CTM wall-clock time is spent in Fast-J using average clouds and 28% when using cloud quadrature. Because the UCI CTM runs a minimalist tropospheric chemistry and a linearized stratospheric chemistry (see Hsu and Prather, 2010), it keeps track of only 32 species. More complete models like Oslo CTM3 (Sovde et al., 2012) and WACCM (Marsh et
al., 2013) calculate transport and chemistry on about 100 species. In CTMs like these the fractional cost of Cloud-J should be only 4-7%. Comparing Solar-J to Fast-J in single-atmosphere tests shows what is expected, Solar-J costs are 3.5x greater because of the much larger number of spectral bands needed for heating (100 vs 18). A minor feature is that cloudy atmospheres cost about 10% more than clear atmospheres because Fast-J inserts extra layers at the top of clouds to enhance the accuracy of the finite-difference equations.

In a series of comparisons on a single-socket multi-threaded CPU, we find that Solar-J takes 5x more wall clock time than RRTMG. This is not surprising given the cost of solving an 8-stream vs. 2-stream RT problem. An additional cost of Solar-J (not included above) is spherical geometry. With RRTMG, 50% of the grid cells are in sunlight and require RT solutions. With Solar-J, however, important photochemistry and solar heating occur in the atmosphere when the surface is past sunset (see Figure 3) involving about 56% of the grid cells, a 12 %
increase in radiatively active grid cells. One could expect that RRTMG will correct this error and end up with similar increase in coverage and cost.

Most climate models, even at the highest resolutions, have individual grid cells with fractional, overlapping cloud layers. Although 3D RT models can be used to solve for the average heating and photolysis rates, most climate models decompose the cloud structures into ICAs, for example, through cloud-resolving models
(Khairoutdinov et al., 2005) or from cloud fractional coverage and a decorrelation distance for overlapping cloud layers (Prather, 2015). The ICAs are horizontally homogeneous and can be solved using the 1D RT codes of RRTM or Solar-J. Comparisons between Solar-J and RRTMG for clouds in Section 3 are done with a single 1D plane-parallel, ICA-like atmosphere (i.e., 100% cloud fraction in each cloudy layer), an idealized case.

Although the different approaches for fractional cloud cover were not directly tested here, it is worth looking at
how Solar-J and RRTMG might treat cloud fields in climate models. The Monte Carlo ICA (McICA, Pincus et al., 2003) method selects both ICAs and spectral intervals randomly in each grid square. Every spectral interval is sampled only once, and each may have a different ICA selected according to its fractional area (frequency of occurrence). With 100+ bins-ICA combinations, the ICAs are well sampled, but there may be instances in which a few, key, large-energy bins are not sampled accurately. The McICA approach when suitably averaged
over time has no mean bias in average heating rates but very large root-mean square (rms) errors: e.g., ±105 W m$^{-2}$ in surface heating with SZA = 45º; ±3 K/day in layers with partly cloudy atmospheres (Pincus et al., 2003). It is cost efficient in that each wavelength bin requires only 1 ICA calculation. Solar-J uses cloud quadrature, introduced by Neu et al. (2007), selecting up to 4 cloud profiles (QCAs) based on total optical depth to represent





four types of atmosphere: mostly clear, typical cirrus clouds, typical stratus clouds, and very thick frontal or
cumulus clouds. While each grid cell may have up to 4 QCAs, on average there are only 2.75. Solar-J then
calculates all wavelength bins using all QCAs to compute the average. Cloud-J (Prather 2015) compared a
number of approximations for calculating average photolysis rates (Js) within a sample of 640 tropical
atmospheres where the number of ICAs per grid cell ranging from 1 to 3,500 and averaged 170. Compared to
the exact answer defined by separate calculations with all the weighted ICAs, cloud quadrature achieves rms
errors in instantaneous cell-averaged Js of 0 to 3% throughout the troposphere, with most levels being 0-1%.
When Cloud-J is run selecting random ICAs (using all wavelengths for each ICA, not the McICA approach), 50
random ICAs (18x more cost) are needed to achieve the accuracy of cloud quadrature.

From the point of view of chemistry-climate models, large rms errors in Js cannot be tolerated because the
chemistry is non-linear and such errors are not likely to average. In climate models, there are threshold
processes, like aerosol and ozone heating preventing cloud formation (e.g., Koch and DelGenio, 2010), for
which heating noise may not simply average out. Errors in heating rates do not always have symmetric
responses in terms of climate (e.g., Hsu et al., 2013). Although Pincus et al. (2003) tested climate forecasting
with an early version of McICA, it is not clear how forecast skill with modern, high-resolution models are
impacted by the biases in RRTMG-SW. Of course, RRTMG could adopt cloud quadrature with 2.8x greater
cost and eliminate most of their rms errors in heating.

All of these standard features of Solar-J (8-streams, spherical geometry, cloud quadrature) increase the
computational cost, but one can argue that the improved fidelity in the solar heating of the atmosphere and
radiative forcing of the climate is worth the cost. The question is what fraction of the total computational cost
of a climate simulation would be used by Solar-J? If we estimate the fractional cost of RRTMG in a full
atmosphere-ocean climate simulation to be 1-3%, then replacing it with Solar-J (5x) and including cloud
quadrature (2.8x), would increase this to 13-39%. At the low-end of this range, the substantially improved and
less noisy physics is probably worth it; but at the upper-end, it is prohibitive. In either case, it is worthwhile to
pursue a range of computer science and algorithmic approaches to reduce these costs as discussed in sections 4.2
and 4.3 below.

## 4.2 Computer science options

A profiling of the Solar-J code shows that the Fast-J core, consisting of scattering matrix generator and block-
tridiagonal solver, is the dominant cost. These two subroutines are already well optimized in terms of single
CPU multi-threading; however, porting Fast-J to computers with graphical processing units (GPUs) has shown
promise for greater speed up. One effort targeted a single GPU and demonstrated speedups via CUDA
(Compute Unified Device Architecture) tuning of ~50x relative to the CPU time if a large number of column
atmospheres (200+) were concurrently evaluated (Artico et al., 2015). Another effort used a field-
programmable gate array (FPGA) with the advantage that it applies to a single column calculation. The FPGA
resulted in ~4x speedup and a rather dramatic 35x energy savings compared to the multicore processor
computation (Rezai et al., 2016). Fast-J was also optimized for the Xeon Phi on the Babbage test platform at
DOE NERSC and achieved ~3x speedups with only a subset of the cores.





Great computational acceleration could be realized with GPU systems when a number of column atmospheres are being simultaneously evaluated. For each grid cell Solar-J calculates about 100 wavelength bins and an average of 2.75 ICAs per grid square. Giving each CPU/GPU node a 3x3 grid cell square (~2,500 column

atmospheres) could achieve 10x or greater speedups for Solar-J and be appropriate for a massively parallel climate simulation (e.g., 32,000 nodes for a 50-km global grid). With such speedups, Solar-J costs would be comparable or possibly less than those of the current RRTMG, and thus become a marginal cost in the climate simulation.

**4.3 Other parameterizations for wavelength bins**

Solar-J uses its own optimization of wavelength bins at ultra-violet and visible wavelengths (0.18 to 0.8 μm), which is based on long experience with $O_2$ and $O_3$ cross sections and the need to calculate accurate J-values. We accept that RRTMG and its parent code RRTM represent current best practice and accuracy in

characterizing the absorption of infrared sunlight (0.8 to 4 μm) in the Earth's atmosphere and have adopted the RRTMG code exactly for bins and all gaseous absorbers. Solar-J's computational cost is clearly driven by the additional 82 infrared bins adopted from RRTMG. Alternative methods of parameterizing these infrared bins needs to be examined: e.g., 14 bins (Chu, 1992; Grant and Grossman, 1998); 34 bins (Fu and Liou, 1992); 36 bins (Cole, 2005). Any of these would result is a 1.5x to 2.5x savings for Solar-J. We recognize that the

infrared bins adopted in RRTMG are based on accurate representation of the line-by-line calculations, and thus adopting these reduced-bin parameterizations will introduce new errors, but further research will be needed to determine whether these errors maybe an acceptable trade-off for speed gain.

Many of these other parameterizations (e.g., Chu, 1992) are based only on water vapor and do not include the other trace gases that that are represented in RRTMG-SW: $O_2$ in the visible and infrared, $CH_4$, and $CO_2$. These

gases add to the complexity of the RRTM model, and thus we investigate their importance in tropospheric heating rates. For our clear-sky case here (Table 4, Figure 2), we find an average tropospheric heating rate of 2.1 K/day. The contribution of $CH_4$ to this total is 0.1%; that of $CO_2$ is complex because of the stratospheric self-shielding but is less that ±1% in the troposphere; and that of $O_2$ is about 3% uniformly throughout the troposphere. If we can find a way of treating the $O_2$ heating separately, then the effort to find an abbreviated

number of spectral intervals can focus on water vapor.

**5 Conclusions**

We present a new solar radiation module designed for accurate, consistent calculation of photolysis rates and heating rates in the atmosphere: Solar-J version 7.5. In a chemistry-climate model, Solar-J supplies the needs of solar heating of the atmosphere and surface, photolysis rates, and photosynthetic activity. Climate models are

increasingly including short-lived gases and aerosols as radiative forcing components, and the accurate simulation of these under different climates requires some level of interactive chemistry and photolysis rates.




The components of Solar-J are chosen to achieve the highest accuracy while still providing a module intended as a standard component of chemistry-climate simulations. From Cloud-J we take the 8-stream scattering model, semi-spherical geometry, ultraviolet transmission, and cloud quadrature. From RRTMG-SW, we take

the detailed spectral intervals for the visible and infrared developed from the RRTM reference code. Solar-J matches RRTMG-SW except where the improved physics leads to more accurate results. Trying to use the best physics for all these components comes with a cost: A simple comparison shows Solar-J costs 5x that of RRTMG-SW. We show that Solar-J can be optimized on GPUs and achieve speeds similar to RRTMG-SW. While this opens up great opportunities for the new generation of high-performance computers, it also

complicates the simple implementation of Solar-J in a climate model.

Solar-J is a starting point. In trying to further increase the simulation fidelity of the interaction of sunlight with the many components of the climate system, we can focus on the three major sources of costs/error (spectral intervals, multi-stream radiative transfer, and complex cloud systems) and, in parallel, on the opportunities for accelerated performance with new computational architectures. Ideally, this is a tradeoff where the community

optimizes computational cost to have comparable errors in all three parameterizations, and, moreover, these parameterization errors in treating solar radiation are clearly mapped onto changes in the climate simulations. For Solar-J the next steps involve some clear improvements: (i) move the S17-S18 boundary to the beginning of the $O_3$ Chappuis absorption near 0.44 μm, and (ii) develop a more realistic and diverse range of cirrus clouds and their optical properties (e.g., Yang et al., 2015). A third opportunity is to test some of the published,

simpler models for water vapor absorption against RRTMG-SW. A larger project will be to put Solar-J into a climate model and evaluate how errors in solar radiation may affect the climate simulations.

## 6 Code availability

The most recent version of Solar-J can be found at anonymous ftp://128.200.14.8/public/junoh/Solar-J/. A complete version of Solar-J code and data, along with some standalone test cases, are included in a zip file as a Supplement to this article.

## 7 Acknowledgement

This work is supported by the U.S. Department of Energy, Office of Science, Biological and Environmental Research Program under Award numbers DE-SC0007021 and DE-SC0012536, and by NASA Modeling, Analysis and Prediction (MAP) Program under award number NNX13AL12G. This research used resources of the National Energy Research Scientific Computing Center, a DOE Office of Science User Facility supported by the Office of Science of the U.S. Department of Energy under Contract No. DE-AC02-05CH11231.




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





Table 1: Some key configuration parameters for Solar-J v7.5 wavelength bins: solar-flux weighted wavelength ($\lambda_{eff}$) within the range between $\lambda_{beg}$ and $\lambda_{end}$, solar fluxes in photons cm$^{-2}$ s$^{-1}$ ($S_{phot}$) and in Wm$^{-2}$ ($S_{watt}$), Rayleigh cross-section ($X_{Rayl}$) and yields for photosynthetically active radiation ($Y_{par}$).

| bin | $\lambda_{eff}$ (μm) | $\lambda_{beg}$ (μm) | $\lambda_{end}$ (μm) | $S_{phot}$ (cm$^{-2}$s$^{-1}$) | $S_{Watt}$ (W m$^{-2}$) | $X_{Rayl}$ (cm$^2$) | $Y_{PAR}$ (/phot) |
|---|---|---|---|---|---|---|---|
| S01 | 0.187 | | | 1.391E+12 | 0.0147 | 5.073E-25 | |
| S02 | 0.191 | | | 1.627E+12 | 0.0168 | 4.479E-25 | |
| S03 | 0.193 | | | 1.664E+12 | 0.0170 | 4.196E-25 | |
| S04 | 0.196 | | | 9.278E+11 | 0.0094 | 3.906E-25 | |
| S05 | 0.202 | | | 7.842E+12 | 0.0766 | 3.355E-25 | |
| S06 | 0.208 | | | 4.680E+12 | 0.0445 | 2.929E-25 | |
| S07 | 0.211 | | | 9.918E+12 | 0.0930 | 2.736E-25 | |
| S08 | 0.214 | | | 1.219E+13 | 0.1128 | 2.581E-25 | |
| S09 | 0.261 | | | 6.364E+14 | 4.818 | 1.049E-25 | |
| S10 | 0.267 | | | 4.049E+14 | 2.962 | 9.492E-26 | |
| S11 | 0.277 | | | 3.150E+14 | 2.218 | 8.103E-26 | |
| S12 | 0.295 | 0.2910 | 0.2982 | 5.893E+14 | 3.703 | 6.135E-26 | |
| S13 | 0.303 | 0.2982 | 0.3074 | 7.670E+14 | 4.670 | 5.424E-26 | |
| S14 | 0.310 | 0.3074 | 0.3124 | 5.041E+14 | 3.063 | 4.925E-26 | |
| S15 | 0.316 | 0.3124 | 0.3203 | 8.895E+14 | 5.414 | 4.516E-26 | |
| S16 | 0.333 | 0.3203 | 0.3450 | 3.852E+15 | 22.28 | 3.644E-26 | 0.0514 |
| S17 | 0.383 | 0.3450 | 0.4124 | 1.547E+16 | 77.17 | 2.082E-26 | 0.4855 |
| S18 | 0.599 | 0.4124 | 0.7780 | 1.805E+17 | 608.68 | 4.427E-27 | 0.6760 |
| S19 | 0.973 | 0.778 | 1.242 | | 349.96 | 5.380E-28 | |
| S20 | 1.267 | 1.242 | 1.299 | | 25.59 | 1.559E-28 | |
| S21 | 1.448 | 1.299 | 1.626 | | 102.96 | 9.578E-29 | |
| S22 | 1.767 | 1.626 | 1.942 | | 56.01 | 4.241E-29 | |
| S23 | 2.039 | 1.942 | 2.151 | | 22.40 | 2.347E-29 | |
| S24 | 2.309 | 2.151 | 2.500 | | 23.50 | 1.441E-29 | |
| S25 | 2.748 | 2.500 | 3.077 | | 20.20 | 7.290E-30 | |
| S26 | 3.404 | 3.077 | 3.846 | | 12.25 | 3.117E-30 | |
| S27 | 5.362 | 3.846 | 12 | | 12.58 | 8.053E-31 | |



*Table 2*

Table 2. Spectral properties for liquid and ice water clouds and stratospheric sulfate aerosols: the real and imaginary refractive indices ($n_r$ and $n_i$), ratio of optical to geometric cross section (Q), single scattering albedo (SSA), and the asymmetry factor (g)

| bin | Liquid water cloud: $r_e$ = 12 μm, ρ = 1.00 g cm$^{-3}$ | | | | | Ice water cloud: $r_e$ = 48 μm, ρ = 0.917 g cm$^{-3}$ | | | | |
|---|---|---|---|---|---|---|---|---|---|---|
| | $n_r$ | $n_i$ | Q | SSA | g | $n_r$ | $n_i$ | Q | SSA | g |
| S12 | 1.350 | 1.8E-08 | 2.054 | 1.0000 | 0.867 | 1.336 | 5.8E-09 | 2.021 | 1.0000 | 0.812 |
| S13 | 1.349 | 1.5E-08 | 2.053 | 1.0000 | 0.869 | 1.333 | 5.4E-09 | 2.021 | 1.0000 | 0.812 |
| S14 | 1.348 | 1.4E-08 | 2.052 | 1.0000 | 0.869 | 1.332 | 5.1E-09 | 2.022 | 1.0000 | 0.812 |
| S15 | 1.347 | 1.3E-08 | 2.055 | 1.0000 | 0.869 | 1.331 | 4.8E-09 | 2.022 | 1.0000 | 0.812 |
| S16 | 1.345 | 9.5E-09 | 2.057 | 1.0000 | 0.869 | 1.328 | 4.3E-09 | 2.023 | 1.0000 | 0.812 |
| S17 | 1.340 | 3.4E-09 | 2.062 | 1.0000 | 0.869 | 1.321 | 3.0E-09 | 2.025 | 1.0000 | 0.812 |
| S18 | 1.333 | 3.1E-08 | 2.089 | 1.0000 | 0.863 | 1.310 | 1.7E-08 | 2.034 | 1.0000 | 0.812 |
| S19 | 1.328 | 2.8E-06 | 2.118 | 0.9996 | 0.858 | 1.302 | 1.7E-06 | 2.047 | 0.9991 | 0.812 |
| S20 | 1.324 | 1.2E-05 | 2.144 | 0.9986 | 0.852 | 1.297 | 1.3E-05 | 2.055 | 0.9946 | 0.812 |
| S21 | 1.321 | 1.6E-04 | 2.155 | 0.9851 | 0.854 | 1.293 | 2.4E-04 | 2.060 | 0.9246 | 0.812 |
| S22 | 1.313 | 3.2E-04 | 2.179 | 0.9752 | 0.852 | 1.284 | 2.2E-04 | 2.069 | 0.9413 | 0.812 |
| S23 | 1.302 | 9.2E-04 | 2.197 | 0.9427 | 0.858 | 1.272 | 1.2E-03 | 2.076 | 0.7876 | 0.812 |
| S24 | 1.283 | 6.7E-04 | 2.220 | 0.9610 | 0.855 | 1.251 | 4.7E-04 | 2.083 | 0.9088 | 0.812 |
| S25 | 1.239 | 1.0E-01 | 2.211 | 0.4979 | 0.970 | 1.125 | 1.0E-01 | 2.071 | 0.5107 | 0.812 |
| S26 | 1.428 | 5.1E-02 | 2.268 | 0.5240 | 0.939 | 1.496 | 1.6E-01 | 2.102 | 0.5408 | 0.812 |
| S27 | 1.317 | 2.2E-02 | 2.409 | 0.6809 | 0.861 | 1.326 | 2.9E-02 | 2.144 | 0.5245 | 0.812 |
| | | | | | | | | | | |
| bin | Strat sulf, volc.: $r_e$ =0.422 μm, ρ=1.69 g cm$^{-3}$ | | | | | Strat sulf, bkgrd: $r_e$ =0.130 μm, ρ=1.69 g cm$^{-3}$ | | | | |
| | $n_r$ | $n_i$ | Q | SSA | g | $n_r$ | $n_i$ | Q | SSA | g |
| S05 | 1.505 | 0.0E+00 | 2.612 | 1.0000 | 0.732 | 1.505 | 0.0E+00 | 2.966 | 1.0000 | 0.698 |
| S06 | 1.505 | 0.0E+00 | 2.638 | 1.0000 | 0.728 | 1.505 | 0.0E+00 | 2.936 | 1.0000 | 0.698 |
| S07 | 1.505 | 0.0E+00 | 2.620 | 1.0000 | 0.735 | 1.505 | 0.0E+00 | 2.919 | 1.0000 | 0.699 |
| S08 | 1.505 | 0.0E+00 | 2.628 | 1.0000 | 0.734 | 1.505 | 0.0E+00 | 2.904 | 1.0000 | 0.699 |
| S09 | 1.472 | 0.0E+00 | 2.604 | 1.0000 | 0.718 | 1.472 | 0.0E+00 | 2.435 | 1.0000 | 0.711 |
| S10 | 1.469 | 0.0E+00 | 2.606 | 1.0000 | 0.710 | 1.469 | 0.0E+00 | 2.379 | 1.0000 | 0.711 |
| S11 | 1.464 | 0.0E+00 | 2.556 | 1.0000 | 0.707 | 1.464 | 0.0E+00 | 2.271 | 1.0000 | 0.711 |
| S12 | 1.456 | 0.0E+00 | 2.500 | 1.0000 | 0.695 | 1.456 | 0.0E+00 | 2.087 | 1.0000 | 0.709 |
| S13 | 1.452 | 0.0E+00 | 2.474 | 1.0000 | 0.690 | 1.452 | 0.0E+00 | 1.998 | 1.0000 | 0.708 |
| S14 | 1.451 | 0.0E+00 | 2.461 | 1.0000 | 0.686 | 1.451 | 0.0E+00 | 1.940 | 1.0000 | 0.706 |
| S15 | 1.451 | 0.0E+00 | 2.449 | 1.0000 | 0.683 | 1.451 | 0.0E+00 | 1.892 | 1.0000 | 0.704 |
| S16 | 1.450 | 0.0E+00 | 2.432 | 1.0000 | 0.676 | 1.450 | 0.0E+00 | 1.766 | 1.0000 | 0.698 |
| S17 | 1.445 | 0.0E+00 | 2.475 | 1.0000 | 0.675 | 1.445 | 0.0E+00 | 1.432 | 1.0000 | 0.683 |
| S18 | 1.431 | 1.7E-08 | 3.017 | 1.0000 | 0.723 | 1.431 | 1.7E-08 | 0.620 | 1.0000 | 0.593 |
| S19 | 1.424 | 1.5E-06 | 2.212 | 1.0000 | 0.663 | 1.424 | 1.5E-06 | 0.193 | 1.0000 | 0.434 |
| S20 | 1.417 | 8.6E-06 | 1.431 | 0.9999 | 0.605 | 1.417 | 8.6E-06 | 0.090 | 0.9998 | 0.336 |
| S21 | 1.430 | 9.4E-05 | 1.173 | 0.9988 | 0.570 | 1.430 | 9.4E-05 | 0.065 | 0.9972 | 0.291 |
| S22 | 1.422 | 4.7E-04 | 0.724 | 0.9910 | 0.511 | 1.422 | 4.7E-04 | 0.033 | 0.9782 | 0.225 |
| S23 | 1.410 | 1.3E-03 | 0.475 | 0.9672 | 0.456 | 1.410 | 1.3E-03 | 0.021 | 0.9184 | 0.182 |
| S24 | 1.388 | 2.1E-03 | 0.305 | 0.9288 | 0.397 | 1.388 | 2.1E-03 | 0.013 | 0.8166 | 0.148 |
| S25 | 1.319 | 5.1E-02 | 0.253 | 0.3855 | 0.302 | 1.319 | 5.1E-02 | 0.040 | 0.0768 | 0.106 |
| S26 | 1.366 | 1.7E-01 | 0.424 | 0.1714 | 0.214 | 1.366 | 1.7E-01 | 0.098 | 0.0219 | 0.074 |
| S27 | 1.406 | 2.1E-01 | 0.274 | 0.0744 | 0.091 | 1.406 | 2.1E-01 | 0.073 | 0.0066 | 0.033 |





*Table 3*

Table 3: Standard tropical atmosphere and the two cloud profiles implemented in both Solar-J and RRTMG. Height (Z) and pressure (P) are edge values; others are layer averages.

| Layer | $Z_{edge}$ (km) | $P_{edge}$ (hPa) | T (K) | $O_3$ ($cm^{-3}$) | $H_2O$ (g/kg) | Stratus Cloud LWC (g m$^{-3}$) | $r_e$ | Cirrus Cloud IWC (g m$^{-3}$) | $r_e$ (μm) |
|---|---|---|---|---|---|---|---|---|---|
| 58 | 75.25 | 0.020 | | | | | | | |
| 57 | 59.58 | 0.200 | 232.4 | 1.27E+09 | 0 | 0 | 0 | 0 | 0 |
| 56 | 54.95 | 0.384 | 242.4 | 1.17E+10 | 0 | 0 | 0 | 0 | 0 |
| 55 | 51.11 | 0.636 | 259.9 | 2.81E+10 | 0 | 0 | 0 | 0 | 0 |
| 54 | 47.91 | 0.956 | 268.1 | 5.79E+10 | 0 | 0 | 0 | 0 | 0 |
| 53 | 45.25 | 1.345 | 266.9 | 1.07E+11 | 0 | 0 | 0 | 0 | 0 |
| 52 | 42.97 | 1.806 | 263.9 | 1.84E+11 | 0 | 0 | 0 | 0 | 0 |
| 51 | 40.97 | 2.348 | 259.9 | 3.01E+11 | 0 | 0 | 0 | 0 | 0 |
| 50 | 39.18 | 2.985 | 255.2 | 4.66E+11 | 0 | 0 | 0 | 0 | 0 |
| 49 | 37.52 | 3.740 | 250.7 | 6.78E+11 | 0 | 0 | 0 | 0 | 0 |
| 48 | 35.96 | 4.646 | 245.1 | 9.63E+11 | 0 | 0 | 0 | 0 | 0 |
| 47 | 34.46 | 5.757 | 240.3 | 1.30E+12 | 0 | 0 | 0 | 0 | 0 |
| 46 | 32.97 | 7.132 | 237.2 | 1.70E+12 | 0 | 0 | 0 | 0 | 0 |
| 45 | 31.50 | 8.837 | 234.3 | 2.20E+12 | 0 | 0 | 0 | 0 | 0 |
| 44 | 30.04 | 10.95 | 231.6 | 2.87E+12 | 0 | 0 | 0 | 0 | 0 |
| 43 | 28.61 | 13.57 | 228.7 | 3.56E+12 | 0 | 0 | 0 | 0 | 0 |
| 42 | 27.20 | 16.81 | 225.2 | 4.24E+12 | 0 | 0 | 0 | 0 | 0 |
| 41 | 25.81 | 20.82 | 221.4 | 4.88E+12 | 0 | 0 | 0 | 0 | 0 |
| 40 | 24.45 | 25.80 | 216.0 | 4.67E+12 | 0 | 0 | 0 | 0 | 0 |
| 39 | 23.12 | 31.96 | 211.9 | 4.36E+12 | 0 | 0 | 0 | 0 | 0 |
| 38 | 21.80 | 39.60 | 211.4 | 3.93E+12 | 0 | 0 | 0 | 0 | 0 |
| 37 | 20.48 | 49.07 | 209.8 | 3.31E+12 | 0 | 0 | 0 | 0 | 0 |
| 36 | 19.25 | 60.18 | 205.9 | 2.01E+12 | 0 | 0 | 0 | 0 | 0 |
| 35 | 18.10 | 73.07 | 202.2 | 1.47E+12 | 0 | 0 | 0 | 0 | 0 |
| 34 | 17.05 | 87.73 | 196.6 | 1.02E+12 | 0 | 0 | 0 | 1.10E-06 | 6.99 |
| 33 | 16.08 | 104.2 | 191.1 | 4.10E+11 | 0 | 0 | 0 | 5.88E-05 | 17.45 |
| 32 | 15.17 | 122.6 | 192.1 | 4.06E+11 | 0.01 | 0 | 0 | 1.32E-04 | 21.03 |
| 31 | 14.28 | 142.8 | 197.6 | 3.25E+11 | 0.02 | 0 | 0 | 3.49E-04 | 26.29 |
| 30 | 13.43 | 165.0 | 203.6 | 3.28E+11 | 0.05 | 0 | 0 | 8.40E-04 | 32.17 |
| 29 | 12.59 | 188.9 | 209.8 | 3.23E+11 | 0.07 | 0 | 0 | 1.02E-03 | 33.66 |
| 28 | 11.78 | 214.6 | 216.6 | 3.45E+11 | 0.15 | 0 | 0 | 1.46E-03 | 36.54 |
| 27 | 11.00 | 242.1 | 223.4 | 3.55E+11 | 0.29 | 0 | 0 | 2.01E-03 | 39.31 |
| 26 | 10.23 | 271.2 | 230.0 | 3.88E+11 | 0.29 | 0 | 0 | 2.19E-03 | 40.12 |
| 25 | 9.48 | 302.1 | 236.2 | 4.29E+11 | 0.75 | 0 | 0 | 3.41E-03 | 44.39 |
| 24 | 8.76 | 334.6 | 242.4 | 4.66E+11 | 0.79 | 0 | 0 | 1.92E-04 | 22.90 |
| 23 | 8.06 | 368.6 | 248.2 | 5.02E+11 | 0.90 | 0 | 0 | 3.35E-04 | 26.03 |
| 22 | 7.38 | 403.9 | 253.4 | 5.40E+11 | 1.90 | 0 | 0 | 1.85E-05 | 13.38 |
| 21 | 6.73 | 440.3 | 258.1 | 5.80E+11 | 1.90 | 0 | 0 | 2.59E-07 | 5.01 |
| 20 | 6.10 | 477.5 | 262.2 | 6.21E+11 | 1.90 | 0 | 0 | 8.58E-08 | 3.89 |
| 19 | 5.51 | 515.4 | 266.0 | 6.22E+11 | 4.07 | 0 | 0 | 2.13E-07 | 4.79 |
| 18 | 4.94 | 553.7 | 269.6 | 6.46E+11 | 4.79 | 0 | 0 | 5.98E-08 | 3.58 |
| 17 | 4.41 | 591.9 | 272.7 | 6.84E+11 | 4.79 | 0 | 0 | 0 | 0 |
| 16 | 3.91 | 629.9 | 275.3 | 7.23E+11 | 4.79 | 0 | 0 | 0 | 0 |
| 15 | 3.44 | 667.2 | 278.0 | 6.80E+11 | 8.14 | 0 | 0 | 0 | 0 |
| 14 | 3.00 | 703.7 | 280.8 | 6.19E+11 | 11.80 | 0 | 0 | 0 | 0 |
| 13 | 2.60 | 738.9 | 282.9 | 6.46E+11 | 11.80 | 0 | 0 | 0 | 0 |
| 12 | 2.22 | 772.7 | 284.9 | 6.72E+11 | 11.80 | 0 | 0 | 0 | 0 |
| 11 | 1.88 | 804.6 | 286.9 | 6.97E+11 | 11.80 | 2.66E-02 | 9.60 | 0 | 0 |
| 10 | 1.57 | 834.6 | 288.6 | 7.20E+11 | 11.80 | 2.05E-02 | 9.60 | 0 | 0 |
| 9 | 1.30 | 862.3 | 290.3 | 7.41E+11 | 11.80 | 8.66E-02 | 9.60 | 0 | 0 |
| 8 | 1.05 | 887.6 | 291.8 | 6.30E+11 | 14.79 | 1.21E-01 | 9.60 | 0 | 0 |
| 7 | 0.83 | 910.3 | 293.0 | 6.22E+11 | 15.30 | 9.67E-02 | 9.60 | 0 | 0 |
| 6 | 0.64 | 930.3 | 294.2 | 6.34E+11 | 15.30 | 4.22E-02 | 9.60 | 0 | 0 |
| 5 | 0.49 | 947.7 | 295.3 | 6.45E+11 | 15.30 | 1.53E-02 | 9.60 | 0 | 0 |
| 4 | 0.35 | 962.3 | 296.1 | 6.54E+11 | 15.30 | 6.62E-03 | 9.60 | 0 | 0 |
| 3 | 0.25 | 974.3 | 296.7 | 6.62E+11 | 15.30 | 3.01E-03 | 9.60 | 0 | 0 |
| 2 | 0.10 | 990.9 | 297.7 | 6.69E+11 | 15.30 | 5.69E-04 | 9.60 | 0 | 0 |



| 1 | 0.00 | 1002.0 | 298.9 | 6.76E+11 | 15.30 | 1.56E-04 | 9.60 | 0 | 0 |

*Table 4*

Table 4. Spectral shortwave radiation energy budget in $Wm^{-2}$ under clear aerosol-free July conditions: Solar-J versus RRTMG. The solar constant is set at 1360.8 $W\ m^{-2}$. For easy comparison, some Solar-J bins are combined to best match RRTMG's band of similar range and vice versa.

**Table 4a.** Clear-Sky Solar Radiation Budget Comparison (W m$^{-2}$)

| Solar-J S-bins | S1-S4 | S5-S9 | S10-S16 | S17 | S18 |
|---|---|---|---|---|---|
| λ(nm) | 177-200 | 200-275 | 275-345 | 345-412 | 412-778 |
| TOA(down) | 0.06 | 5.14 | 44.31 | 77.17 | 608.68 |
| TOA(up) | 0.00 | 0.01 | 7.52 | 16.89 | 54.23 |
| Atmosphere | 0.06 | 5.14 | 18.01 | 0.05 | 32.32 |
| *-Stratosphere* | *0.06* | *5.14* | *16.97* | *0.04* | *9.41* |
| *-Troposphere* | *0.00* | *0.00* | *1.04* | *0.01* | *22.91* |
| Surface | 0.00 | 0.00 | 18.78 | 60.23 | 522.13 |

| RRTMG Bands | / | B28 | B27 | B26 | B25+B24 |
|---|---|---|---|---|---|
| λ(nm) | / | 200-263 | 263-345 | 345-442 | 442-778 |
| TOA(down) | / | 3.06 | 49.88 | 128.79 | 562.34 |
| TOA(up) | / | 0.02 | 7.37 | 25.75 | 50.30 |
| Atmosphere | / | 3.05 | 23.22 | 0.00 | 29.70 |
| *-Stratosphere* | / | *3.04* | *22.11* | *0.00* | *8.60* |
| *-Troposphere* | / | *0.01* | *1.11* | *0.00* | *21.10* |
| Surface | / | 0.00 | 19.29 | 103.04 | 482.33 |

**Table 4b.** Clear-Sky Solar Radiation Budget Comparison (W m$^{-2}$)

| Solar-J S-bins | S19 | S20 | S21 | S22 | S23 |
|---|---|---|---|---|---|
| λ(μm) | 0.78-1.24 | 1.24-1.30 | 1.30-1.63 | 1.63-1.94 | 1.94-2.15 |
| TOA(down) | 349.96 | 25.59 | 102.96 | 56.01 | 22.40 |
| TOA(up) | 15.29 | 1.28 | 2.18 | 1.43 | 0.6 |
| Atmosphere | 87.85 | 2.36 | 60.44 | 29.36 | 9.52 |
| *-Stratosphere* | *0.00* | *0.12* | *0.29* | *0.20* | *0.41* |
| *-Troposphere* | *87.85* | *2.23* | *60.15* | *29.16* | *9.11* |
| Surface | 246.83 | 21.96 | 40.35 | 25.22 | 12.28 |

| RRTMG Bands | B23 | B22 | B21 | B20 | B19 |
|---|---|---|---|---|---|
| λ(μm) | 0.78-1.24 | 1.24-1.30 | 1.30-1.63 | 1.63-1.94 | 1.94-2.15 |
| TOA(down) | 343.86 | 24.16 | 102.37 | 55.32 | 22.31 |
| TOA(up) | 14.90 | 1.20 | 2.03 | 1.40 | 0.57 |
| Atmosphere | 86.49 | 2.23 | 61.64 | 29.01 | 9.56 |
| *-Stratosphere* | *0.00* | *0.12* | *0.30* | *0.20* | *0.42* |
| *-Troposphere* | *86.49* | *2.11* | *61.34* | *28.84* | *9.15* |
| Surface | 242.47 | 20.74 | 38.71 | 24.91 | 12.18 |

**Table 4c.** Clear-Sky Solar Radiation Budget Comparison (W m$^{-2}$)

| Solar-J S-bins | S24 | S25 | S26 | S27 | All bands |
|---|---|---|---|---|---|
| λ(nm) | 2.15-2.50 | 2.50-3.08 | 3.08-3.85 | 3.85-12 | 0.18-12 |
| TOA(down) | 23.50 | 20.20 | 12.25 | 12.58 | 1360.80 |
| TOA(up) | 0.75 | 0.00 | 0.19 | 0.06 | 100.43 |
| Atmosphere | 8.28 | 20.17 | 7.28 | 10.35 | 291.13 |
| *-Stratosphere* | *0.07* | *1.65* | *0.17* | *1.28* | *35.80* |
| *-Troposphere* | *8.21* | *18.52* | *7.11* | *9.07* | *255.33* |
| Surface | 14.48 | 0.03 | 4.78 | 2.18 | 969.24 |

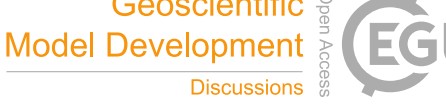



| RRTMG Bands | B18 | B17 | B16 | B29 | All bands |
|---|---|---|---|---|---|
| λ(nm) | 2.15-2.50 | 2.50-3.08 | 3.08-3.85 | 3.85-12 | 0.20-12 |
| TOA(down) | 23.60 | 20.25 | 12.04 | 12.82 | 1360.80 |
| TOA(up) | 0.76 | 0.00 | 0.18 | 0.05 | 104.54 |
| Atmosphere | 7.89 | 20.22 | 7.16 | 10.55 | 290.70 |
| *-Stratosphere* | *0.06* | *1.66* | *0.16* | *1.30* | *37.91* |
| *-Troposphere* | *7.84* | *18.56* | *7.01* | *9.26* | *252.79* |
| Surface | 14.95 | 0.03 | 4.70 | 2.22 | 965.55 |

*Table 5*

Table 5. Comparison of Solar-J and RRTMG for top-of-atmosphere (TOA), atmosphere, and surface radiation budgets (W m$^{-2}$) across four SZAs. Also shown is the cloud radiative effect (CRE) of a typical marine stratus cloud, for which the atmospheric absorption is split into above-cloud, in-cloud, and below-cloud.

| SZA | 0° | | 21° | | 62° | | 84° | |
|---|---|---|---|---|---|---|---|---|
| Flux (Wm$^{-2}$) | 1360.8 | | 1268.4 | | 634.2 | | 149.1 | |
| **Clear-Sky Radiation Budget (W m$^{-2}$)** | | | | | | | | |
| | Solar-J | RRTMG | Solar-J | RRTMG | Solar-J | RRTMG | Solar-J | RRTMG |
| TOA(up) | 100.4 | 104.5 | 96.1 | 100.2 | 63.9 | 67.2 | 28.0 | 28.4 |
| Atmosphere | 291.0 | 290.7 | 276.7 | 276.0 | 166.2 | 164.6 | 56.8 | 55.4 |
| Surface | 969.2 | 965.6 | 895.7 | 892.3 | 404.1 | 402.4 | 64.2 | 65.3 |
| **Cloud Radiative Effect of a Marine Stratus Cloud (Wm$^{-2}$)** | | | | | | | | |
| | Solar-J | RRTMG | Solar-J | RRTMG | Solar-J | RRTMG | Solar-J | RRTMG |
| TOA | +469.2 | +454.7 | +447.7 | +436.6 | +258.9 | +252.0 | +50.6 | +48.8 |
| Atmosphere | +91.0 | +91.5 | +80.9 | +81.7 | +24.0 | +22.1 | -1.5 | -1.9 |
| *Above-cloud* | *+23.6* | *+26.8* | *+20.0* | *+25.5* | *+12.4* | *+13.1* | *+3.2* | *+1.6* |
| *In-cloud* | *+75.5* | *+71.7* | *+68.8* | *+63.1* | *+17.1* | *+13.9* | *-2.9* | *-2.1* |
| *Below-cloud* | *-8.1* | *-6.9* | *-7.9* | *-6.7* | *-5.5* | *-4.9* | *-1.6* | *-1.4* |
| Surface | -560.2 | -546.2 | -528.6 | -518.2 | -283.0 | -274.1 | -49.1 | -47.0 |





*Table 6*

Table 6. Cirrus ice cloud optical properties: total optical depth $\tau$ for Solar-J and $\delta$-scaled $\tau'$ for RRTMG, asymmetry factor g, and absorption optical depth, $\tau_{abs}$ for bins S18 to S27. See Table 1 for wavelength ranges and RRTMG-equivalent bins.

| Sbins | S18 | S19 | S20 | S21 | S22 | S23 | S24 | S25 | S26 | S27 |
|---|---|---|---|---|---|---|---|---|---|---|
| $\lambda_{eff}$ | 599nm | 973nm | 1.27μm | 1.45μm | 1.77μm | 2.04μm | 2.31μm | 2.75μm | 3.40μm | 5.36μm |
| **Total Optical Depth ($\tau$ for Solar-J and reduced $\tau'$ for RRTMG schemes)** | | | | | | | | | | |
| Solar-J | 0.4287 | 0.4322 | 0.4345 | 0.4360 | 0.4383 | 0.4404 | 0.4425 | 0.4380 | 0.4470 | 0.4591 |
| EC92 | 0.2488 | 0.2462 | 0.2462 | 0.2385 | 0.2385 | 0.2276 | 0.2276 | 0.3313 | 0.3313 | 0.3313 |
| Fu96 | 0.1535 | 0.1581 | 0.1563 | 0.1627 | 0.1640 | 0.1575 | 0.1932 | 0.3709 | 0.3382 | 0.3177 |
| Key02 | 0.0923 | 0.0943 | 0.0950 | 0.1041 | 0.1032 | 0.1277 | 0.1065 | 0.1783 | 0.2159 | 0.2266 |
| **Asymmetry Factor, g = $A_1$/3** | | | | | | | | | | |
| Solar-J | 0.7643 | 0.7642 | 0.7641 | 0.7640 | 0.7639 | 0.7639 | 0.7638 | 0.7639 | 0.7635 | 0.7631 |
| EC92 | 0.4406 | 0.4425 | 0.4425 | 0.4484 | 0.4484 | 0.4579 | 0.4579 | 0.4907 | 0.4907 | 0.4907 |
| Fu96 | 0.4591 | 0.4680 | 0.4803 | 0.4987 | 0.5168 | 0.5670 | 0.5870 | 0.6744 | 0.3411 | 0.0000 |
| Key02 | 0.4694 | 0.4692 | 0.4691 | 0.4707 | 0.4707 | 0.4757 | 0.4731 | 0.4866 | 0.4807 | 0.4858 |
| **Total absorbing optical depth ($\tau_{abs}$)** | | | | | | | | | | |
| Solar-J | 0.0000 | 0.0003 | 0.0018 | 0.0257 | 0.0201 | 0.0758 | 0.0317 | 0.2163 | 0.2075 | 0.2126 |
| EC92 | 0.0000 | 0.0004 | 0.0004 | 0.0247 | 0.0247 | 0.0558 | 0.0558 | 0.3063 | 0.3063 | 0.3063 |
| Fu96 | 0.0000 | 0.0003 | 0.0022 | 0.0232 | 0.0231 | 0.0743 | 0.0289 | 0.1294 | 0.1743 | 0.1972 |
| Key02 | 0.0000 | 0.0001 | 0.0011 | 0.0178 | 0.0162 | 0.0619 | 0.0290 | 0.1491 | 0.1788 | 0.1983 |



*Figure 1*

Figure 1. Solar-J extends Fast-J's solar wavelength bands by combining and modifying RRTMG's band 24 and 25 (442-778 nm) and adopts all RRTMG's bands longwards of 778 nm (see text for detail).





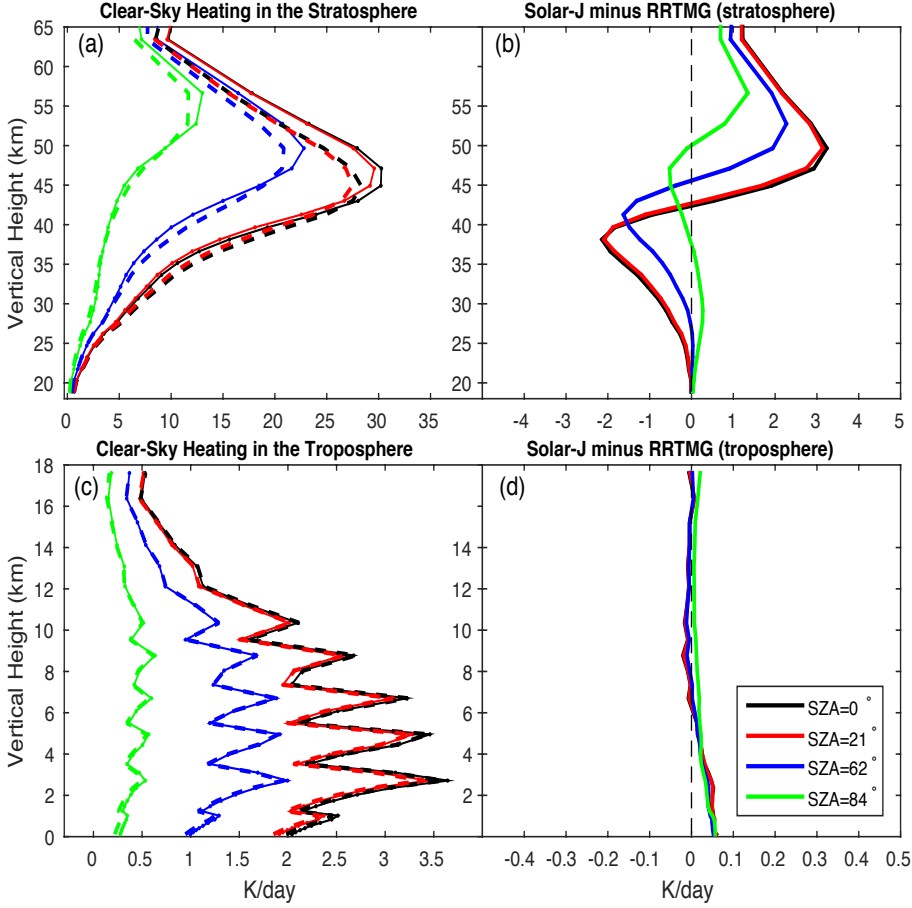

*Figure 2*

Figure 2. Aerosol-free cloudless atmospheric heating profiles of Solar-J (solid lines) and RRTMG (dashed lines) and the difference, Solar-J minus RRTMG, for a typical July tropical atmosphere at 4 solar zenith angles with Lambertian surface albedo = 0.06 (left and right sides). The plot is further split into stratosphere and troposphere (top and bottom rows). Note that the scale of the x-axis, K/day, is 10 times larger for the stratosphere.





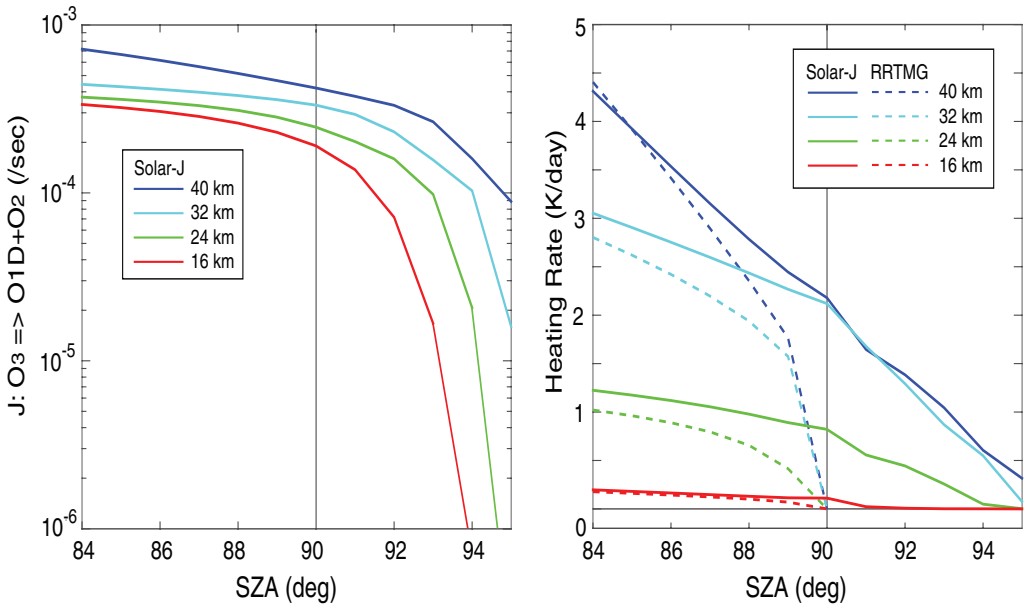

*Figure 3*

Figure 3. Ozone photolysis rates ($J_{O3}$) from Solar-J (left panel) and the corresponding atmospheric heating rates under clear sky (right panel) from Solar-J (solid lines) and RRTMG (dashed lines) for large solar zenith angles at 4 different altitudes. RRTMG's heating rates reduce to zeros at SZA= 90∘ due to the lack of sphericity correction in the plane-parallel approximation; whereas the impact of sphericity on the direct solar beam path is included in Solar-J.





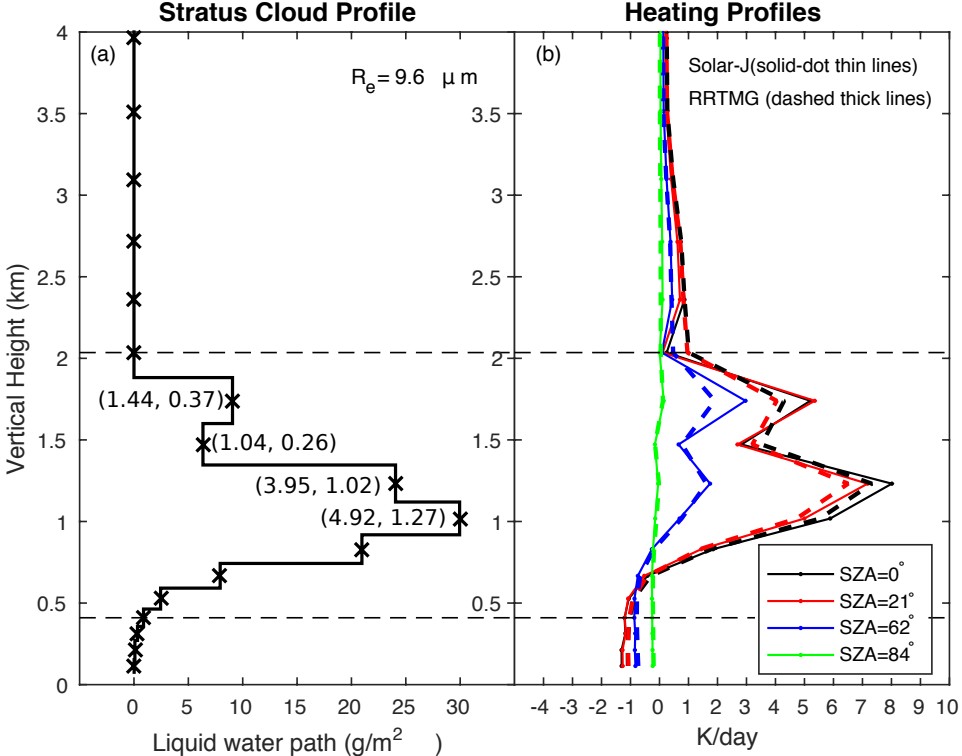

*Figure 4*

Figure 4. (a) Marine stratus cloud profile in terms of liquid water path (LWP, g m$^{-2}$) and effective radius ($r_e$,

μm). The optical depth and δ-scaled optical depth ($\tau$, $\tau^{'}$) are shown in parentheses for the top five cloud layers.

(b) Cloud heating profiles from Solar-J (solid lines) and RRTM (dashed lines) at fours SZAs.

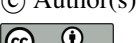

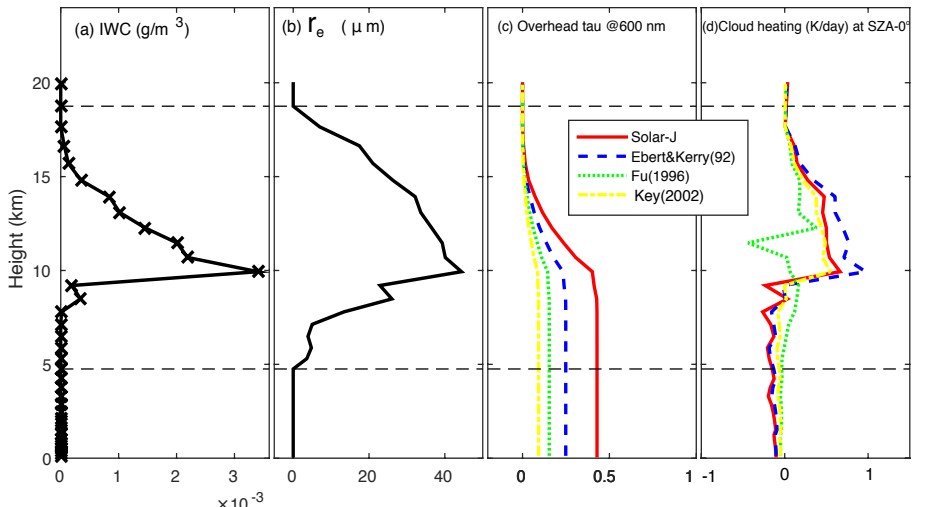

*Figure 5*

Figure 5. Profiles of (a) ice water content (IWC, g m$^{-3}$) and (b) effective radius ($r_e$, μm) as prescribed for both Solar-J and RRTMG. The in-cloud region, about 4-18 km, is enclosed by two horizontal dashed lines. (c) Profiles of cumulative optical depth τ at 600 nm from Solar-J and from the 3 RRTMG parameterizations for which τ is δ-scaled. (d) Cirrus cloud heating rate profiles (K/day) at SZA=0°.





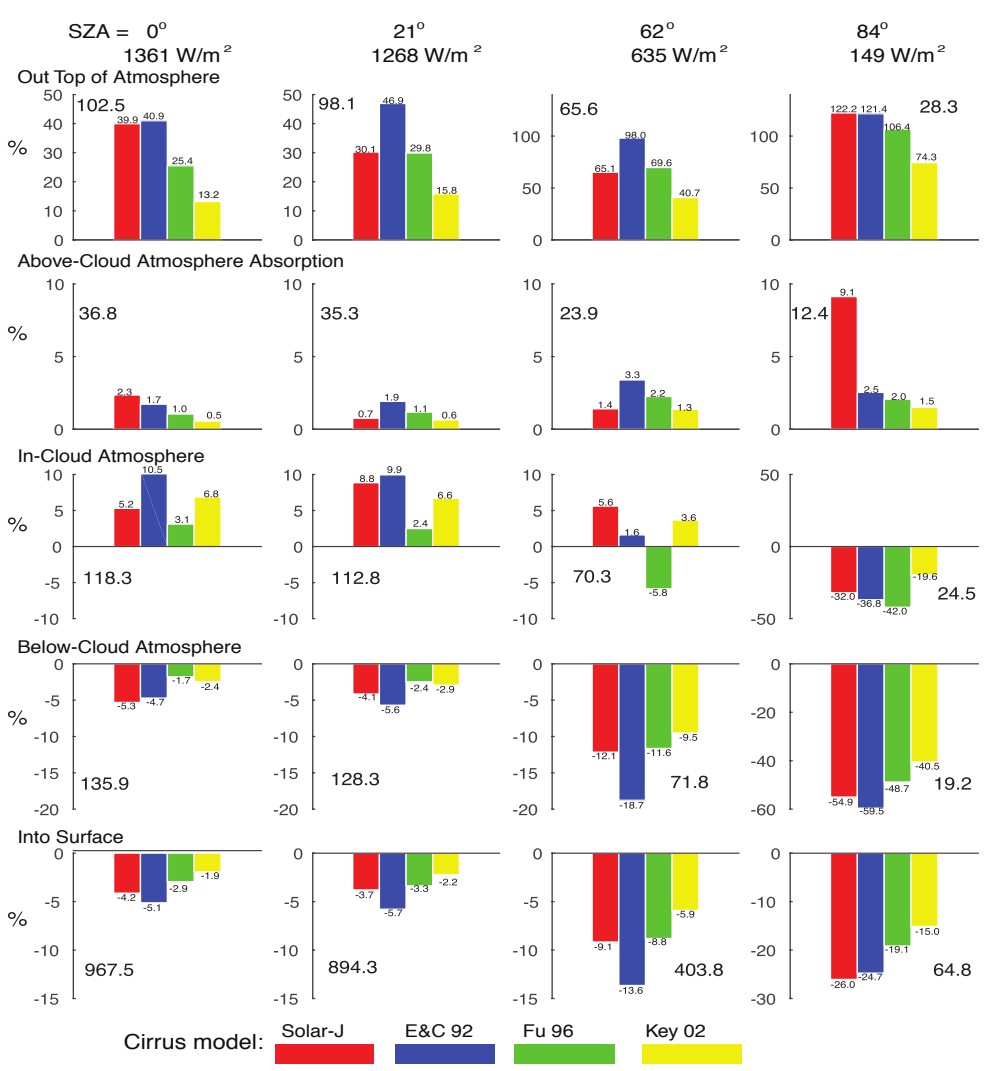

*Figure 6*

Figure 6. Percent changes (%) in shortwave radiation energy budget relative to the aerosol-free clear sky (surface albedo = 0.06) caused by a cirrus cloud using four different models: Solar-J and the three RRTMG parametrizations for ice clouds. Results are shown for 4 different solar zenith angles. Changes in the vertical column are divided into 5 regions: top of atmosphere, atmospheres above, within and below the cirrus cloud, and at the surface. Single numbers in bold shown in the corner of each panel are the clear-sky energy budget in W m$^{-2}$ averaged over Solar-J and RRTMG for each region. Percentage changes are also shown in text at the end of each color bar. Note that different y-axis scales have been used for large SZAs.