# Peer review of "A Radiative Transfer Module for Calculating Photolysis Rates and Solar Heating in Climate Models: Solar-J 7.5"

_Geoscientific Model Development, 2017_

## Referee Comment (RC1) · Anonymous Referee #1 · 24 Mar 2017

This is a well written paper describing original research of high significance. I recommend the publication of the manuscript in its current form.

---

## Referee Comment (RC2) · Anonymous Referee #2 · 11 Apr 2017

This paper is generally very well written providing substantial details on methods and scientific explanation of comparative results, and limitations. It addresses needs for a more complete fast radiative transfer model for climate (and forecast) models by extending the Fast-J/Cloud-J code mainly by extending infrared spectral bin coverage to 12 microns following the RRTMG-SW model. The resulting model, called Solar-J, appears to combine the best features of both packages.

I recommend the publication of the manuscript following minor revisions.

P.S. Access to Solar-J worked fine.

Scientific presentation:

[Figure]

1. While this referee has some gaps regarding the scientific background of all aspects involved in this work, the scientific content, descriptions, and justifications appear well done and very detailed and extensive. This is very commendable. No needed revisions have been identified on that front.

2. Minor revisions are needed regarding the presentation of Solar-J in the context of its inheritance from Fast-J, Fast-J2 and Cloud-J. In the conclusion section, it is indicated that 8-stream scattering, semi-spherical geometry, UV transmission, and cloud quadrature were taken from Cloud-J. This referee did not find any clear prior mention that the 8-stream and semi-spherical geometry were already present in the original code from which Solar-J began. Actually, the 8- stream code would have been from Fast-J2 while the spherical earth consideration for solar radiation would have been in Fast-J. There is much discussion in the introduction and elsewhere on the merit of the 8-stream approach, giving the initial impression that this is a new added feature, i.e. while it was present in Fast-J2.

There are back and forth references to Fast-J and Cloud-J which tends to confuse if one was not already familiar with both. It would be best to present the contributions and relations of each to Solar-J in the introduction (e.g. in the second paragraph of the introduction) and, from that point, maybe just refer to Cloud-J afterwards which would have been the starting point of Solar-J.

For clarity, it would be needed to indicate, from the beginning, likely in the introduction since the 8-stream approach is highlighted here, all components and features stemming from Fast-J and its successors Fast-J2 and Cloud-J, prior to the additions made to generate Solar-J. It is acknowledged that this is done regarding the spectral configuration and also, but only later, the Cloud-J cloud quadrature.

The mention of both 2-stream and 8-stream is provided in the introduction to highlight (it needs to be made explicit in the introduction) the advantage of "continuing with the 8-stream approach from Cloud-J" vs. the 2-stream approach of RRTMG-SW.

Considering the above comments, there might be some benefit in correspondingly revising the introduction (and related text locations here and there such as the abstract – see (3) below).

3. The quality of the abstract and conclusions section is not as high generally as that of the remainder of the paper. Comments on the composition of the conclusions section is provided later. A few comments mostly on the scientific presentation of the abstract follows bellow.

- As alluded to in (2) above, there is mention of Solar-J including the 8-stream scattering a few other features without indicating that these were inherited from Fast-J/Cloud-J. The sentence ' Solar-J is a . . .' in lines 15-16 could moved to line 11 with mention of Cloud-J. The following sentence needs to then attribute these mentioned features as inherited from Cloud-J.

- A new paragraph could then begin from line 17 indicating the extension based on RRTMG-SW.

- Lines 18-19. The statement "successfully matches RRTMG's atmospheric heating profile" does not seem consistent with Fig 2b unless this is meant to refer to the general features of Fig 2a in addition to F2c,d. One might consider adding another comment pointing to the level of difference (e.g. "with maximum differences of 3 K in the upper stratosphere stemming from the absence of radiation below 200 microns and the coarse UV-bin resolution of RRMTG-SW")

- Lines 23-24. "less systematic" is unclear – to remove if not clarified. Meaning of "larger" is also unclear.

- Line 24-25. There a missing link between the previous sentence referring to discrepancies/differences for the cirrus cloud example (not indicating if Solar-J is better) and the following sentence referring to Solar-J combining the best of both models. Maybe this has to do with the phrasing of the previous sentence (referring to lines 23-24).

- Line 28 and also line 367 (conclusion). "about 5x" is for clear-sky only. Another 2.8x should be indicated the cloud quadrature (see page 14).

3. Line 352. For completeness, may be best to indicate "biases relative to Solar-J results" or something like "differences relative to Solar-J results identified here are errors caused by the 2-stream approximation used with RRTMG.")

4. There is referencing to applicability of Solar-J with climate and chemistry-climate models. This should/could be extended at least to weather prediction models (if not also air quality models as well - CTMs and Coupled chemistry-weather).

Composition corrections and suggestions:

- Both RRTMG vs RRTMG-SW are used in the abstract and various places in this work. If the short version RRTMG is preferred, would be best to indicate, in the introduction, that it will be used from that point onwards. In the abstract, likely best to just use RRTMG-SW.

- Line 39. Suggest replacing "The major" by "Major" since other major challenges are also present as indicated later in the introduction.

- Line 45. Replacing ", however" by ". However"

- Line 46. "Thus" unnecessary. Can start with "In terms of . . ."

- Line 269. Replace "into stratosphere" by "into the stratosphere"

- Line 280. Suggested rephrasing. "The benefit of moving the Solar-J (and Cloud-J) band edge to 442 nm should be investigated."

- Line 329. Should "Figure 3(a)" be indicated as "Figure 3a" for consistency with this paper or is it the labelling used in Painemal et al. (2016).

- Line 358. Suggest "One other source"

- Line 369. The mention of EC92, Fu96 and Key02 should be accompanied here be

the references (which are present in the reference list).

- Line 417. Might be good to replace in this proportionality" by "similarly proportional".

- Line 444. Might be best to replace "Fast-J 8-stream" to "Solar-J 8 stream"

- Line 446. "Feautrier solves" to "Feautrier approach solves".

- Line 454. "Cloud-J (and hence Fast-J) has"

- Line 471. "increases"

- Lines 484-485. Suggestion of adding commas: "approach, when suitably averaged over time, "

- Line 489. "of atmospheres"

- Line 493. Rephrasing needed for "where the number of ICAs per grid cell ranging from 1 to 3,500 and averaged 170."

- Line 504. Suggest removing the sentence beginning with "Obviously".

- Line 517. "of a scattering matric generator and a block-" (added "a"s)

- Line 524. "in a ∼4x"

- Line 537. Rephrase "on long experience"

- Line 549. "that that are" to "that are"

- Line 553. "less that" to "less than"

- Conclusions section. Composition to use past tense (presented vs present, taken vs was taken, "can focus vs focused, . . .). A revisit of the conclusion composition and content might be beneficial.

- Line 557. "accurate, consistent with . . . in the atmosphere" seem too much and not precise. Suggest instead finding a sentence or two referring to Solar-J incorporating

strengths of both Cloud-J and RRTMG-SW.

- Line 562. Would be better to replace "The components of" by referring to Solar-J combining the best of Cloud-J and RRTMG-SW.

- Line 574-575. Rephrasing needed with "Ideally, there is a tradeoff . . . in all three parameterizations." - Line 575. " and, however, these .." to ". These . . .."

- Line 576. " are clearly mapped" to "would be mapped" or "would have some impact on climate. .." or "are expected to impact. . .." or "could have an impact ..."

- Lines 577-578. "For Solar-J, the next steps consist of (i) moving the . . . and (ii) developing .." or something similar.

- Line 579. Wonder if "A third opportunity" could be rephrased.

Tables: Moving the table captions outside the table frames will likely be necessary.

Figures:

- Figure 4. (1) Extra space in Fig 4a x-axis title units (g/mˆ2 ) to remove. (2) Missing space in Fig 2b legend at the top "Solar-J(solid. . .". (3) In caption, replace "at fours SZAs" by "at four SZAs"

- Figure 5. (1) In legend of Fig 5c,d replace "Ebert&Kerry" by "Ebert&Curry". (2) Missing space in Fig 5d title "(d)Cloud. . ."

- Figure 6. Numbers below above colour bars in panels seem rather small. However, not much space to increase their size. So maybe ok.

References: I did not check that all references are accounted for.

---

## Referee Comment (RC3) · Anonymous Referee #2 · 11 Apr 2017

It appears that a significant part of the introduction deals with scattering while section section two deals does not refer to it as much and focusses on other aspects. For improved balance, it might be worthwhile to consider moving some discussion on scattering from the introduction to section two and dwell more on the range of aspects to be covered in section two (and some background on the motivation) in the introduction.

---

## Author Comment (AC1) · 25 Apr 2017

*The authors wish to thank the two anonymous reviewers for their encouraging comments. Below is our response (blue, Italic) to each reviewer's comments. We have improved the presentation following reviewer #2's suggestions, particularly improving on the presentation in the abstract, introduction and conclusions. The revised manuscript with tracked changes is also included.*

Reviewer #1
This is a well written paper describing original research of high significance. I recommend the publication of the manuscript in its current form.

*Thanks for the encouragement. We hope that this work will indeed have a long-term impact on climate modeling.*

Reviewer #2

This paper is generally very well written providing substantial details on methods and scientific explanation of comparative results, and limitations. It addresses needs for a more complete fast radiative transfer model for climate (and forecast) models by extending the Fast-J/Cloud-J code mainly by extending infrared spectral bin coverage to 12 microns following the RRTMG-SW model. The resulting model, called Solar-J, appears to combine the best features of both packages. I recommend the publication of the manuscript following minor revisions. P.S. Access to Solar-J worked fine.

*We thank for the positive, encouraging, detailed, and constructive review. We have revised the manuscript per your suggestions below and provided a point-by-point response to each comment.*

Scientific presentation:
1. While this referee has some gaps regarding the scientific background of all aspects involved in this work, the scientific content, descriptions, and justifications appear well done and very detailed and extensive. This is very commendable. No needed revisions have been identified on that front.
*Thanks. Good to hear!*

2. Minor revisions are needed regarding the presentation of Solar-J in the context of its inheritance from Fast-J, Fast-J2 and Cloud-J. In the conclusion section, it is indicated that 8-stream scattering, semi-spherical geometry, UV transmission, and cloud quadrature were taken from Cloud-J. This referee did not find any clear prior mention that the 8- stream and semi-spherical geometry were already present in the original code from which Solar-J began. Actually, the 8- stream code would have been from Fast-J2 while the spherical earth consideration for solar radiation would have been in Fast-J. There is much discussion in the introduction and elsewhere on the merit of the 8-stream approach, giving the initial impression that this is a new added feature, i.e. while it was present in Fast-J2.

There are back and forth references to Fast-J and Cloud-J which tends to confuse if one was not already familiar with both. It would be best to present the contributions and

relations of each to Solar-J in the introduction (e.g. in the second paragraph of the introduction) and, from that point, maybe just refer to Cloud-J afterwards which would have been the starting point of Solar-J.

For clarity, it would be needed to indicate, from the beginning, likely in the introduction since the 8-stream approach is highlighted here, all components and features stemming from Fast-J and its successors Fast-J2 and Cloud-J, prior to the additions made to generate Solar-J. It is acknowledged that this is done regarding the spectral configuration and also, but only later, the Cloud-J cloud quadrature. The mention of both 2-stream and 8-stream is provided in the introduction to highlight (it needs to be made explicit in the introduction) the advantage of "continuing with the 8-stream approach from Cloud-J" vs. the 2-stream approach of RRTMG-SW.
The mention of both 2-stream and 8-stream is provided in the introduction to highlight (it needs to be made explicit in the introduction) the advantage of "continuing with the 8-stream approach from Cloud-J" vs. the 2-stream approach of RRTMG-SW.

 Considering the above comments, there might be some benefit in correspondingly revising the introduction (and related text locations here and there such as the abstract – see (3) below).

*Thanks. We agree.  We have rewritten the 2[nd] paragraph of the introduction in the revised manuscript to include a comprehensive overview of the development and relationship between the work done previously known as Fast-J, Fast-J2, Fast- JX and Cloud-J.  We more carefully explain what Cloud-J entails.  And we made it clear that Cloud-J will serve as the starting point of Solar-J.   We also went through the manuscript to replace Fast-J with Cloud-J.*

3. The quality of the abstract and conclusions section is not as high generally as that of the remainder of the paper. Comments on the composition of the conclusions section is provided later. A few comments mostly on the scientific presentation of the abstract follows bellow. - As alluded to in (2) above, there is mention of Solar-J including the 8-stream scattering a few other features without indicating that these were inherited from Fast-J/Cloud-J. The sentence ' Solar-J is a . . .' in lines 15-16 could moved to line 11 with mention of Cloud-J. The following sentence needs to then attribute these mentioned features as inherited from Cloud-J. - A new paragraph could then begin from line 17 indicating the extension based on RRTMG-SW. - Lines 18-19.

*Thanks. We have followed your advice by moving the sentence from Line 15-16. to Line 11 and beginning a new paragraph starting on Line 17.  We further have edited the abstract for clarity based on the restructure.*

The statement "successfully matches RRTMG's atmospheric heating profile" does not seem consistent with Fig 2b unless this is meant to refer to the general features of Fig 2a in addition to F2c,d. One might consider adding another comment pointing to the level of difference (e.g. "with maximum differences of 3 K in the upper stratosphere stemming from the absence of radiation below 200 microns and the coarse UV-bin resolution of RRMTG-SW")
*That is a good point. We only meant tropospheric. We edited "atmospheric heating profiles" to "tropospheric heating profiles".  The stratospheric heating difference is caused by the different methods in simulating the UV absorption of $O_2$ and $O_3$. We only adopted RRTMG-SW's gas absorption bins from the infrared range.*

- Lines 23-24. "less systematic" is unclear – to remove if not clarified. Meaning of "larger" is also unclear.

*We now indicate the magnitude, "15-50% depending on…".*

 - Line 24-25. There a missing link between the previous sentence referring to discrepancies/differences for the cirrus cloud example (not indicating if Solar-J is better) and the following sentence referring to Solar-J combining the best of both models. Maybe this has to do with the phrasing of the previous sentence (referring to lines 23-24)
*We did not mean to link these two sentences, and we now break between these sentences to begin a new paragraph.*

- Line 28 and also line 367 (conclusion). "about 5x" is for clear-sky only. Another 2.8x should be indicated the cloud quadrature (see page 14).

*We realize now that there might be a misunderstanding on this part. The 2.8x additional cost from using cloud quadrature scheme is relative to a single atmosphere in Solar-J, not relative to RRTMG-SW's cloud overlapping scheme in parallel. We don't know the costs of RRTMG-SW if their McICA is used, but presume it would be similar to using a single atmosphere. We now make it clear in the conclusion in Lines 583-584: "A simple comparison shows the cost of Solar-J is 5x that of RRTMG-SW for a single atmosphere, and if the cloud quadrature scheme for overlapping cloud fields (Neu et al., 2007; Prather, 2015) is applied to either code, the cost increases additionally by 2.8x". In the abstract, we added, "for a single atmosphere".*

3. Line 352. For completeness, may be best to indicate "biases relative to Solar-J results" or something like "differences relative to Solar-J results identified here are errors caused by the 2-stream approximation used with RRTMG.")  *We believe the usage of "bias" in referring to these specific RRTMG-SW results is well justified, not only by the principal of physics (8-stream by nature is more accurate than 2-stream), but also the low bias has been documented in the literature (e.g. Li et al., 2015).*

4. There is referencing to applicability of Solar-J with climate and chemistry-climate models. This should/could be extended at least to weather prediction models (if not also air quality models as well - CTMs and Coupled chemistry-weather).
*Good. "Weather" is included in the revised introduction on Line 71.*

Composition corrections and suggestions:

 - Both RRTMG vs RRTMG-SW are used in the abstract and various places in this work. If the short version RRTMG is preferred, would be best to indicate, in the introduction, that it will be used from that point onwards. In the abstract, likely best to just use RRTMG-SW.
*Agreed. We mostly use RRTMG-SW throughout the text. There are places that we use RRTMG for easy reading. We now added a sentence on Line 103 in the introduction, "In this paper, we use RRTMG as shorthand for RRTMG-SW "*

- Line 39. Suggest replacing "The major" by "Major" since other major challenges are also present as indicated later in the introduction. *fixed.*
- Line 45. Replacing ", however" by ". However" *fixed.*
- Line 46. "Thus" unnecessary. Can start with "In terms of . . ." *fixed.*
- Line 269. Replace "into stratosphere" by "into the stratosphere" *fixed.*
- Line 280. Suggested rephrasing. "The benefit of moving the Solar-J (and Cloud-J) band edge to 442 nm should be investigated." *fixed.*
- Line 329. Should "Figure 3(a)" be indicated as "Figure 3a" for consistency with this paper or is it the labelling used in Painemal et al. (2016). *fixed.*

- Line 358. Suggest "One other source" *fixed.*
- Line 369. The mention of EC92, Fu96 and Key02 should be accompanied here be the references (which are present in the reference list). *The notation of EC92, Fu96 and Key02 are introduced in Section 2.3, Lines 232-234, "For ice clouds three different parameterization are available, and all are tested here (Ebert and Curry, 1992, henceforth EC92; Key, 2002, henceforth Key02; Fu, 1996, henceforth Fu96)."*

- Line 417. Might be good to replace in this proportionality" by "similarly proportional".
*fixed.*
- Line 444. Might be best to replace "Fast-J 8-stream" to "Solar-J 8 stream" *fixed.*
- Line 446. "Feautrier solves" to "Feautrier approach solves". *fixed.*
- Line 454. "Cloud-J (and hence Fast-J) has" *fixed.*
- Line 471. "increases" *fixed.*
- Lines 484-485. Suggestion of adding commas: "approach, when suitably averaged over time, *fixed.*
- Line 489. "of atmospheres" *fixed.*
- Line 493. Rephrasing needed for "where the number of ICAs per grid cell ranging from 1 to 3,500 and averaged 170." *Rephrased.*
- Line 504. Suggest removing the sentence beginning with "Obviously". *You meant "Of course" on Line 504. We changed it to "Alternatively".*
- Line 517. "of a scattering matric generator and a block-" (added "a"s) *fixed.*
- Line 524. "in a ~4x" *fixed.*
- Line 537. Rephrase "on long experience"
 *Changed to "on previously established experience"*

- Line 549. "that that are" to "that are" *fixed*
- Line 553. "less that" to "less than *fixed.*

- Line 574-575. Rephrasing needed with "Ideally, there is a tradeoff . . . in all three " –
*Yes, we got rid of the 'trade off' and simplified the sentence.*

Conclusions section. Composition to use past tense (presented vs present, taken vs was taken, "can focus vs focused, . . .). A revisit of the conclusion composition and content might be beneficial.

*Thanks for the suggestion and the wakeup call.  Which tense to use when describing science in a paper is a longstanding issue. We have traditionally kept to the present tense in the paper where possible unless referring clearly to past work. Most importantly, we needed to proof and check that our usage is consistent.*

 - Line 557. "accurate, consistent with . . . in the atmosphere" seem too much and not precise.

*Have toned it down. Rephrased.*

Suggest instead finding a sentence or two referring to Solar-J incorporating strengths of both Cloud-J and RRTMG-SW.

*Yes, agreed. It is now in the first sentence.*

 - Line 562. Would be better to replace "The components of" by referring to Solar-J combining the best of Cloud-J and RRTMG-SW parameterizations."

*Again, this is mentioned in the first sentence. We have revised this sentence slightly because it did not fit well before, as the reviewer found.*

- Line 575. " and, however, these .." to ". These . . .."

*Fixed.*

- Line 576. " are clearly mapped" to "would be mapped" or "would have some impact on climate. .." or "are expected to impact. . .." or "could have an impact ..."

*This ending section was clearly awkward, and we have redrafted it to be simpler, hopefully clearer, and maintain the original intent.*

- Lines 577-578. "For Solar-J, the next steps consist of (i) moving the . . . and (ii) developing .." or something similar.

*Fixed.*

- Line 579. Wonder if "A third opportunity" could be rephrased.

*Yes. Replaced with "another inquiry"*

 Tables: Moving the table captions outside the table frames will likely be necessary.

*Moved.*

 Figures:
- Figure 4.
 (1) Extra space in Fig 4a x-axis title units (g/m^2 ) to remove. *Fixed.*
 (2) Missing space in Fig 2b legend at the top "Solar-J(solid. . .". *Fixed.*
 (3) In caption, replace "at fours SZAs" by "at four SZAs*" Fixed.*
- Figure 5.
 (1) In legend of Fig 5c,d replace "Ebert&Kerry" by "Ebert&Curry". *Fixed.*

(2) Missing space in Fig 5d title "(d)Cloud. . ." - Figure 6. Numbers below above colour bars in panels seem rather small. However, not much space to increase their size. So maybe ok. *The figure is vectorized and can be blown up If it remains in .eps format in GMD publication.*

References: I did not check that all references are accounted for. *We checked again, thoroughly.*

Additional Comment from Reviewer #2

It appears that a significant part of the introduction deals with scattering while section two deals does not refer to it as much and focusses on other aspects. For improved balance, it might be worthwhile to consider moving some discussion on scattering from the introduction to section two and dwell more on the range of aspects to be covered in section two (and some background on the motivation) in the introduction.
*Thanks for the suggestion. It is a good idea, but we would rather keep the discussion of scattering in the introduction to provide a strong motivation for the community to migrate to a multi-stream algorithm. Rearrangements like that suggested were tried, but it became more awkward and disjointed. Section 2 is already the longest section in this manuscript. Its purpose is to provide enough details for any party whom wants to reproduce our results and can do so.*

[revised manuscript text omitted]

---

## Author Response (AR2)

During the process of responding to the reviews and re-examining our data sets for submission as supplementary material, we have discovered that the vertical water vapor profile in the reference atmosphere was mistakenly averaged over 2 km blocks. This blocky profile is the cause for the zig-zag shapes of the clear sky heating profiles in Fig. 2(c) and the erroneous cirrus heating profile with Fu's parameterization in Fig. 5(d).

We have now adopted a profile from the ECMWF forecast fields that is similar to the old one – but without the averaging - and redone all of our calculations for Solar-J and RRTMG-SW. We have updated all numbers (Tables 3, 4 and 5) and figures (Fig. 2, 4, 5 and 6). The basic results – the differences between RRTMG and Solar-J – were unchanged as were the major conclusions. The only change is that the paper became simpler because we were able to strike out discussion of the zig-zag shapes and the Fu' cirrus heating profiles. Fu's parameterization works as it should be. Overall, there is much better consistency.

This correction to the water vapor profile has negligible effect on the clear-sky radiative budget for each spectral band ($<0.1$ $Wm^{-2}$; Tables 4). However, there is a slight vertical re-arrangement of solar energy absorbed by $H_2O$ since there is more water vapor at the surface but less in the mid-troposphere for the new profile. The impact on the stratus cloud radiative forcing under 1360 $Wm^{-2}$ solar flux at SZA=$0^{o}$, for example, is about 4 $Wm^{-2}$ more reflected sunlight, 3 $Wm^{-2}$ less absorbed in the atmosphere and 1 $Wm^{-2}$ less reaching to the surface (Table 5). This does not impact the comparison of Solar-J and RRTMG-SW since the same degree of correction applies both to RRTMG-SW and Solar-J, for all cases. Nevertheless, we have gone through the manuscript carefully to update all the tables, figures and text.

Please find our point-by-point reply to the reviewers in the pages below, which is followed by tracked changes of the manuscript. We have extensively revised the abstract, introduction and conclusion to make the presentation better per Reviewer #2's comments.

*The authors wish to thank the two anonymous reviewers for their encouraging comments. Below is our response (blue, Italic) to each reviewer's comments. We have improved the presentation following reviewer #2's suggestions, particularly improving on the presentation in the abstract, introduction and conclusions. The revised manuscript with tracked changes is also included.*

Reviewer #1

This is a well written paper describing original research of high significance. I recommend the publication of the manuscript in its current form.

*Thanks for the encouragement. We hope that this work will indeed have a long-term impact on climate modeling.*

Reviewer #2

This paper is generally very well written providing substantial details on methods and scientific explanation of comparative results, and limitations. It addresses needs for a more complete fast radiative transfer model for climate (and forecast) models by extending the Fast-J/Cloud-J code mainly by extending infrared spectral bin coverage to 12 microns following the RRTMG-SW model. The resulting model, called Solar-J, appears to combine the best features of both packages. I recommend the publication of the manuscript following minor revisions. P.S. Access to Solar-J worked fine.

*We thank for the positive, encouraging, detailed, and constructive review. We have revised the manuscript per your suggestions below and provided a point-by-point response to each comment.*

Scientific presentation:
1. While this referee has some gaps regarding the scientific background of all aspects involved in this work, the scientific content, descriptions, and justifications appear well done and very detailed and extensive. This is very commendable. No needed revisions have been identified on that front.

Thanks.

2. Minor revisions are needed regarding the presentation of Solar-J in the context of its inheritance from Fast-J, Fast-J2 and Cloud-J. In the conclusion section, it is indicated that 8-stream scattering, semi-spherical geometry, UV transmission, and cloud quadrature were taken from Cloud-J. This referee did not find any clear prior mention that the 8- stream and semi-spherical geometry were already present in the original code from which Solar-J began. Actually, the 8- stream code would have been from Fast-J2 while the spherical earth consideration for solar radiation would have been in Fast-J. There is much discussion in the introduction and elsewhere on the merit of the 8-stream approach, giving the initial impression that this is a new added feature, i.e. while it was present in Fast-J2.

There are back and forth references to Fast-J and Cloud-J which tends to confuse if one was not already familiar with both. It would be best to present the contributions and relations of each to Solar-J in the introduction (e.g. in the second paragraph of the introduction) and, from that point, maybe just refer to Cloud-J afterwards which would have been the starting point of Solar-J.

For clarity, it would be needed to indicate, from the beginning, likely in the introduction since the 8-stream approach is highlighted here, all components and features stemming from Fast-J and its successors Fast-J2 and Cloud-J, prior to the additions made to generate Solar-J. It is acknowledged that this is done regarding the spectral configuration and also, but only later, the Cloud-J cloud quadrature. The mention of both 2-stream and 8-stream is provided in the introduction to highlight (it needs to be made explicit in the introduction) the advantage of "continuing with the 8-stream approach from Cloud-J" vs. the 2-stream approach of RRTMG-SW.

The mention of both 2-stream and 8-stream is provided in the introduction to highlight (it needs to be made explicit in the introduction) the advantage of "continuing with the 8- stream approach from Cloud-J" vs. the 2-stream approach of RRTMG-SW.

Considering the above comments, there might be some benefit in correspondingly revising the introduction (and related text locations here and there such as the abstract – see (3) below).

Thanks. We agree. We have rewritten the 2$^{nd}$ paragraph of the introduction in the revised manuscript to include a comprehensive overview of the development and relationship between the work done previously known as Fast-J, Fast-J2, Fast- JX and Cloud-J. We more carefully explain what Cloud-J entails. And we made it clear that Cloud-J will serve as the starting point of Solar-J. We also went through the manuscript to replace Fast-J with Cloud-J.

3. The quality of the abstract and conclusions section is not as high generally as that of the remainder of the paper. Comments on the composition of the conclusions section is provided later. A few comments mostly on the scientific presentation of the abstract follows bellow. - As alluded to in (2) above, there is mention of Solar-J including the 8-stream scattering a few other features without indicating that these were inherited from Fast-J/Cloud-J. The sentence 'Solar-J is a . . .' in lines 15-16 could moved to line 11 with mention of Cloud-J. The following sentence needs to then attribute these mentioned features as inherited from Cloud-J. - A new paragraph could then begin from line 17 indicating the extension based on RRTMG-SW. - Lines 18-19.

Thanks. We have followed your advice by moving the sentence from Line 15-16. to Line 11 and beginning a new paragraph starting on Line 17. We further have edited the abstract for clarity based on the restructure.

The statement "successfully matches RRTMG's atmospheric heating profile" does not seem consistent with Fig 2b unless this is meant to refer to the general features of Fig 2a in addition to F2c,d. One might consider adding another comment pointing to the level of difference (e.g. "with maximum differences of 3 K in the upper stratosphere stemming from the absence of radiation below 200 microns and the coarse UV-bin resolution of RRMTG-SW")

That is a good point. We only meant tropospheric. We edited "atmospheric heating profiles" to "tropospheric heating profiles". The stratospheric heating difference is caused by the different methods in simulating the UV absorption of $O_2$ and $O_3$. We only adopted RRTMG-SW's gas absorption bins from the infrared range.

- Lines 23-24. "less systematic" is unclear – to remove if not clarified. Meaning of "larger" is also unclear.

We now indicate the magnitude, "about 20-40 % depending on…".

- Line 24-25. There a missing link between the previous sentence referring to discrepancies/differences for the cirrus cloud example (not indicating if Solar-J is better) and the following sentence referring to Solar-J combining the best of both models. Maybe this has to do with the phrasing of the previous sentence (referring to lines 23-24)

We did not mean to link these two sentences, and we now break between these sentences to begin a new paragraph.

- Line 28 and also line 367 (conclusion). "about 5x" is for clear-sky only. Another 2.8x should be indicated the cloud quadrature (see page 14).

We realize now that there might be a misunderstanding on this part. The 2.8x additional cost from using cloud quadrature scheme is relative to a single atmosphere in Solar-J, not relative to RRTMG-SW's cloud overlapping scheme in parallel. We don't know the costs of RRTMG-SW if their McICA is used, but presume it would be similar to using a single atmosphere. We now make it clear in the conclusion in Lines 583-584: "A simple comparison shows the cost of Solar-J is 5x that of RRTMG-SW for a single atmosphere, and if the cloud quadrature scheme for overlapping cloud fields (Neu et al., 2007; Prather, 2015) is applied to either code, the cost increases additionally by 2.8x". In the abstract, we added, "for a single atmosphere".

3. Line 352. For completeness, may be best to indicate "biases relative to Solar-J results" or something like "differences relative to Solar-J results identified here are errors caused by the 2-stream approximation used with RRTMG.")

We believe the usage of "bias" in referring to these specific RRTMG-SW results is well justified, not only by the principal of physics (8-stream by nature is more accurate than 2-stream), but also the low bias has been documented in the literature (e.g. Li et al., 2015).

4. There is referencing to applicability of Solar-J with climate and chemistry-climate models. This should/could be extended at least to weather prediction models (if not also air quality models as well - CTMs and coupled chemistry- weather).

Good. "Weather" is included in the revised introduction on Line 71.

Composition corrections and suggestions:

- Both RRTMG vs RRTMG-SW are used in the abstract and various places in this work. If the short version RRTMG is preferred, would be best to indicate, in the introduction, that it will be used from that point onwards. In the abstract, likely best to just use RRTMG-SW.

Agreed. We mostly use RRTMG-SW throughout the text. There are places that we use RRTMG for easy reading. We now added a sentence on Line 103 in the introduction, "In this paper, we use RRTMG as shorthand for RRTMG-SW "

- Line 39. Suggest replacing "The major" by "Major" since other major challenges are also present as indicated later in the introduction. fixed.
- Line 45. Replacing ", however" by ". However" fixed.
- Line 46. "Thus" unnecessary. Can start with "In terms of . . ." fixed.
- Line 269. Replace "into stratosphere" by "into the stratosphere" fixed.
- Line 280. Suggested rephrasing. "The benefit of moving the Solar-J (and Cloud-J) band edge to 442 nm should be investigated." fixed.
- Line 329. Should "Figure 3(a)" be indicated as "Figure 3a" for consistency with this paper or is it the labelling used in Painemal et al. (2016). fixed.
- Line 358. Suggest "One other source" fixed.

- Line 369. The mention of EC92, Fu96 and Key02 should be accompanied here be the references (which are present in the reference list).
The notation of EC92, Fu96 and Key02 are introduced in Section 2.3, Lines 232-234, "For ice clouds three different parameterization are available, and all are tested here (Ebert and Curry, 1992, henceforth EC92; Key, 2002, henceforth Key02; Fu, 1996, henceforth Fu96)."

- Line 417. Might be good to replace in this proportionality" by "similarly proportional".
fixed.
- Line 444. Might be best to replace "Fast-J 8-stream" to "Solar-J 8 stream" fixed.
- Line 446. "Feautrier solves" to "Feautrier approach solves". fixed.
- Line 454. "Cloud-J (and hence Fast-J) has" fixed.
- Line 471. "increases" fixed.
- Lines 484-485. Suggestion of adding commas: "approach, when suitably averaged over time,
fixed.
- Line 489. "of atmospheres" fixed.
- Line 493. Rephrasing needed for "where the number of ICAs per grid cell ranging from 1 to 3,500 and averaged 170." Rephrased.
- Line 504. Suggest removing the sentence beginning with "Obviously". You meant "Of course" on Line 504. We changed it to "Alternatively".
- Line 517. "of a scattering matric generator and a block-" (added "a"s) fixed.
- Line 524. "in a ~4x" fixed.
- Line 537. Rephrase "on long experience"
Changed to "on previously established experience"

- Line 549. "that that are" to "that are" fixed
- Line 553. "less that" to "less than fixed.

- Line 574-575. Rephrasing needed with "Ideally, there is a tradeoff . . . in all three " –
Yes, we got rid of the 'trade off' and simplified the sentence.
Conclusions section. Composition to use past tense (presented vs present, taken vs was taken, "can focus vs focused, . . .). A revisit of the conclusion composition and content might be beneficial.

Thanks for the suggestion and the wakeup call. Which tense to use when describing science in a paper is a longstanding issue. We have traditionally kept to the present tense in the paper where possible unless referring clearly to past work. Most importantly, we needed to proof and check that our usage is consistent.

- Line 557. "accurate, consistent with . . . in the atmosphere" seem too much and not precise.

Have toned it down. Rephrased.

Suggest instead finding a sentence or two referring to Solar-J incorporating strengths of both Cloud-J and RRTMG-SW.

Yes, agreed. It is now in the first sentence.

- Line 562. Would be better to replace "The components of" by referring to Solar-J combining the best of Cloud-J and RRTMG-SW parameterizations."

Again, this is mentioned in the first sentence. We have revised this sentence slightly because it did not fit well before, as the reviewer found.

- Line 575. " and, however, these .." to ". These . . .."  Fixed.

- Line 576. " are clearly mapped" to "would be mapped" or "would have some impact on climate. .." or "are expected to impact. . .." or "could have an impact ..."

This ending section was clearly awkward, and we have redrafted it to be simpler, hopefully clearer, and maintain the original intent.

- Lines 577-578. "For Solar-J, the next steps consist of (i) moving the . . . and (ii) developing .." or something similar.  Fixed.
- Line 579. Wonder if "A third opportunity" could be rephrased.  Yes. Replaced with "another inquiry"

Tables: Moving the table captions outside the table frames will likely be necessary. Moved.

Figures:
- Figure 4.
(1) Extra space in Fig 4a x-axis title units (g/m^2 ) to remove.  Fixed.
(2) Missing space in Fig 2b legend at the top "Solar-J(solid. . .".  Fixed.
(3) In caption, replace "at fours SZAs" by "at four SZAs"  Fixed.
- Figure 5.
(1) In legend of Fig 5c,d replace "Ebert&Kerry" by "Ebert&Curry".  Fixed.
(2) Missing space in Fig 5d title "(d)Cloud. . ." - Figure 6. Numbers below above colour bars in panels seem rather small. However, not much space to increase their size. So maybe ok.

The small-font numbers provide backup for the bar charts that need ot be seen together on one page to identify the similarities and differences across models and atmospheric regions.  The figure is vectorized and can be blown up If it remains in .eps format in GMD publication.

References: I did not check that all references are accounted for.  We checked again, thoroughly.

Additional Comment from Reviewer #2

It appears that a significant part of the introduction deals with scattering while section two deals does not refer to it as much and focusses on other aspects. For improved balance, it might be worthwhile to consider moving some discussion on scattering from the introduction to section two and dwell more on the range of aspects to be covered in section two (and some background on the motivation) in the introduction.

Thanks for the suggestion. It is a good idea, but  we would rather keep the discussion of scattering in the introduction to provide a strong motivation for the community to migrate to a multi-stream algorithm. Rearrangements like that suggested were tried, but it became more awkward and disjointed.  Section 2 is already the longest section in this manuscript. Its purpose is to provide enough details for any party whom wants to reproduce our results and can do so.

[revised manuscript text omitted]

Formatted Table

| Page 26: [5] Formatted | JH | 5/25/17 11:00:00 AM |

Font:+Theme Headings Asian (Times New Roman)

| Page 26: [6] Formatted | JH | 5/25/17 11:00:00 AM |

Font:+Theme Headings Asian (Times New Roman)

| Page 26: [7] Deleted | JH | 5/25/17 11:00:00 AM |

Table 3: Standard tropical atmosphere and the two cloud profiles implemented in both Solar-J and RRTMG. Height (Z) and pressure (P) are edge values; others are layer averages.

| Page 26: [8] Formatted | JH | 5/25/17 11:00:00 AM |

Font:+Theme Headings Asian (Times New Roman)

| Page 26: [9] Formatted | JH | 5/25/17 11:00:00 AM |

Font:+Theme Headings Asian (Times New Roman)

| Page 27: [10] Formatted | JH | 5/25/17 11:00:00 AM |

Font:+Theme Headings Asian (Times New Roman)

| Page 27: [11] Formatted | JH | 5/25/17 11:00:00 AM |

Font:+Theme Headings Asian (Times New Roman)

| Page 27: [12] Deleted | JH | 5/25/17 11:00:00 AM |

*Table 4*

Table 4. Spectral shortwave radiation energy budget in $Wm^{-2}$ under clear aerosol-free July conditions: Solar-J versus RRTMG. The solar constant is set at 1360.8 W $m^{-2}$. For easy comparison, some Solar-J bins are combined to best match RRTMG's band of similar range and vice versa.

| Page 28: [13] Formatted Table | JH | 5/25/17 11:00:00 AM |

Formatted Table

| Page 28: [14] Formatted Table | JH | 5/25/17 11:00:00 AM |

Formatted Table

| Page 29: [15] Formatted | JH | 5/25/17 11:00:00 AM |

Line spacing: multiple 1.08 li

| Page 29: [16] Formatted Table | JH | 5/25/17 11:00:00 AM |

Formatted Table

| Page 29: [17] Deleted | JH | 5/25/17 11:00:00 AM |

Table 5. Comparison of Solar-J and RRTMG for top-of-atmosphere (TOA), atmosphere, and surface radiation budgets (W $m^{-2}$) across four SZAs. Also shown is the cloud radiative effect (CRE) of a typical marine stratus cloud, for which the atmospheric absorption is split into above-cloud, in-cloud, and below-cloud.

| Page 30: [18] Deleted | JH | 5/25/17 11:00:00 AM |

*Table 6*

Table 6. Cirrus ice cloud optical properties: total optical depth $\tau$ for Solar-J and $\delta$-scaled $\tau'$ for RRTMG, asymmetry factor g, and absorption optical depth, $\tau_{abs}$ for bins S18 to S27. See Table 1 for wavelength ranges and RRTMG-equivalent bins.

| Page 31: [19] Deleted | JH | 5/25/17 11:00:00 AM |

**Solar-J spectrum: Merging Fast-J and RRTMG**

| | | | |
|---|---|---|---|
| **Fast-J** (Wild et. al, 2000) | Bins 1-17 (177-412 nm) | Bin-18 (412-850 nm) | |
| **Cloud-J** (Prather, 2015) | Bins 1-17 (177-412 nm) | Bin-18 (412-778 nm) | |
| **Solar-J (S-bins)** (this study) | Bins 1-17 (177 - 412 nm) | Bin-18[a] (412-778 nm) Fast-J's $O_3$ (+ weak $O_2+H_2O$) (1 sub-bin) / Bin-18[b] (442-778 nm) Surface $H_2O+O_2$ (4 sub-bins) | Bins 19-27 (778-12195 nm) (78 sub-bins) |
| **RRTMG** (Mlawer et. al, 1997) | Band No. 26-28 (200- 442 nm) (20 sub-bins) | Bands 24&25 (442-778 nm) (14 sub-bins) | Bands 16-23+ Band 29 (778-12195 nm) (78 sub-bins) |
* * *
Page 31: [20] Deleted     JH     5/25/17 11:00:00 AM

*Figure 1*

Page 32: [21] Deleted     JH     5/25/17 11:00:00 AM

[Figure]

Figure 2.

| Page 32: [22] Deleted | JH | 5/25/17 11:00:00 AM |
|---|---|---|

| Page 33: [23] Deleted | JH | 5/25/17 11:00:00 AM |
|---|---|---|

[Figure]

| Page 33: [24] Deleted | JH | 5/25/17 11:00:00 AM |

*Figure 3.*

| Page 34: [25] Deleted | JH | 5/25/17 11:00:00 AM |

[Figure]

| Page 35: [26] Deleted | JH | 5/25/17 11:00:00 AM |

*Figure 5.*

| Page 36: [27] Deleted | JH | 5/25/17 11:00:00 AM |

[Figure]

*Figure 6*